# INFLUENCE DYNAMICS AND STAGEWISE DATA ATTRIBUTION

**Jin Hwa Lee**[=]
University College London
jin.lee.22@ucl.ac.uk

**Matthew Smith**[=]
Independent
matt.ja.smith@gmail.com

**Maxwell Adam**
University of Melbourne
Timaeus
max@timaeus.co

**Jesse Hoogland**
Timaeus
jesse@timaeus.co

## ABSTRACT

Current training data attribution (TDA) methods treat the influence one sample has on another as static, but neural networks learn in distinct stages that exhibit changing patterns of influence. In this work, we introduce a framework for stagewise data attribution grounded in singular learning theory. We predict that influence can change non-monotonically, including sign flips and sharp peaks at developmental transitions. We first validate these predictions analytically and empirically in a toy model, showing that dynamic shifts in influence directly map to the model's progressive learning of a semantic hierarchy. Finally, we demonstrate these phenomena at scale in language models, where token-level influence changes align with known developmental stages.

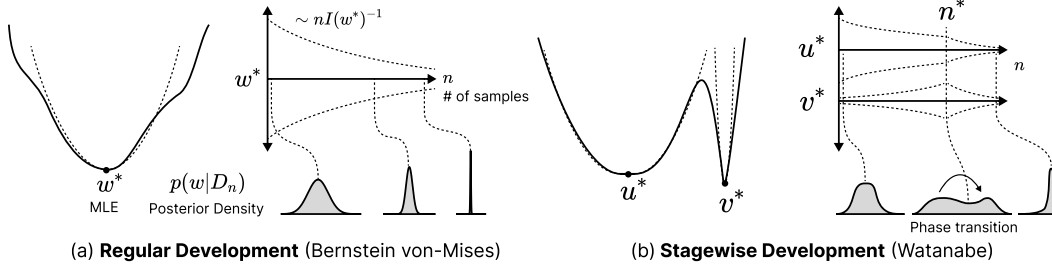

(a) **Regular Development** (Bernstein von-Mises)     (b) **Stagewise Development** (Watanabe)

Figure 1: **Stagewise learning requires stagewise data attribution.** (a) In regular models, development is a uniform, monotonic process of posterior concentration around a single solution (Bernstein–von Mises). (b) In singular models, development is a stagewise process where the posterior undergoes phase transitions (Watanabe's singular learning theory). This stagewise development means the influence one sample has on another can profoundly change over the course of the learning process.

## 1 INTRODUCTION

Training data attribution (TDA) studies how training data shapes model behavior, a central problem in AI interpretability and safety (Cheng et al., 2025; Lehalleur et al., 2025). Understanding attribution requires accounting for the role of stagewise development and learning dynamics: *when* a model encounters a given sample affects *how* the model learns from that sample. What helps the model learn "dog" in early training may actively harm its ability to distinguish "poodle" from "terrier" later.

Currently, however, most approaches to TDA still ignore the role of development. In particular, influence functions (IFs) assume that data ordering has no effect on influence, which implies that

---

= Equal contribution.

influence is static and global over the course of training (Cook, 1977; Cook & Weisberg, 1980). This perspective, inherited from the analysis of regular statistical models, breaks down catastrophically for deep neural networks (see Section 2.1).

The cause of this breakdown is degeneracy: neural networks have degenerate loss landscapes with non-isolated critical points and non-invertible Hessians. Singular learning theory (SLT) predicts that this degeneracy gives rise to stagewise development, where models undergo phase transitions marked by changes in degeneracy and Hessian rank (Watanabe, 2009; 2018). Taken together, this suggests that data attribution should take account of stagewise development, which motivates our work to connect influence functions and developmental phase transitions.

**Contributions.**    We put forth three contributions:

1. **A developmental framework for influence**: We introduce a theoretical framework, grounded in Singular Learning Theory (SLT), that connects influence functions to stagewise development. This predicts that influence is not static but can change non-monotonically (including sign flips and sharp peaks) at the phase transitions that define stagewise learning. This motivates a shift to **stagewise data attribution** that studies the dynamics of influence over time.

2. **Validation in a controlled system**: We analytically derive and empirically confirm our predictions in a hierarchical feature-learning model (Saxe et al., 2019a). We provide a concrete proof-of-concept showing that dynamic shifts in influence directly correspond to the model's sequential learning of the data's hierarchical structure.

3. **Application at scale in language models**: We demonstrate that these developmental phenomena are observable at scale in language models. In particular, we show that influence functions for tokens with key structural roles (e.g., delimiters, induction patterns) undergo non-monotonic, sudden changes that align with known transitions.

Our findings challenge the foundational assumptions of the static influence paradigm, providing a new framework for understanding not just which data points matter, but *when* and *why* they matter during the learning process.

## 2 THEORY

This section develops our theoretical framework for stagewise data attribution. In Section 2.1, we review the theory of stagewise development according to singular learning theory (SLT). We then, in Section 2.2, re-evaluate influence functions through this developmental lens, which motivates a shift from the global, point-wise approach of classical influence functions to a local, distributional variant. Finally, in Section 2.3, we derive our central predictions: influence is not a fixed property but changes non-monotonically, peaking at the phase transitions that define stagewise learning.

### 2.1 FROM UNIFORM TO STAGEWISE DEVELOPMENT

**Developmental Interpretability: from SGD to Bayes and back again.**    While stochastic optimizers (such as SGD and Adam) are the de facto approach to training deep learning systems, their complex dynamics make direct theoretical analysis difficult. To make progress, we follow the recipe of Developmental Interpretability (Lehalleur et al., 2025; Wang et al., 2025b): we model the optimizer's learning trajectory with an idealized Bayesian learning process, then apply singular learning theory (SLT; Watanabe 2009) to make predictions about stagewise development, and finally test those predictions empirically in real networks trained via stochastic optimization.

**The regular learning process (Bernstein–von Mises).**    In *regular* statistical models (with a unique MLE and invertible Fisher information matrix, FIM), the Bernstein–von Mises (BvM) theorem predicts a smooth, monotonic learning process where the posterior narrows around a single solution (Van der Vaart, 2000). More precisely, as the number of samples $n$ increases, the Bayesian posterior converges to a Gaussian centered at the minimum $\boldsymbol{w}^*$ of the population loss with covariance $(n\boldsymbol{I}(\boldsymbol{w}^*))^{-1}$, where $\boldsymbol{I}(\boldsymbol{w}^*)$ is the FIM.

**The singular learning process (Watanabe).** Neural networks violate the regularity assumptions required for the Bernstein–von Mises theorem to hold. Not only do they have no unique minimum, but the loss landscape is *degenerate*: the Fisher information matrix is not everywhere invertible. SLT provides a framework for studying these *singular* models. Watanabe (2009) showed that degeneracy can give rise to stagewise learning, where neural networks undergo a succession of phase transitions between qualitatively distinct solutions, see Figure 1.

In this framework, development is driven by a competition between data fit (or empirical loss, $L_n(\boldsymbol{w}) = \sum_i \ell_i(\boldsymbol{w})$ over a dataset $\mathcal{D}$ of $n$ samples) and model complexity (as measured via a measure of degeneracy known as the local learning coefficient, $\lambda(\boldsymbol{w})$). This evolving tradeoff can lead to first-order phase transitions, where the model abruptly shifts from concentrating in one region to another, and which can change the model's generalization behavior (see Appendix A for a formal treatment). As we will show in Section 2.3, these transitions are also responsible for the non-monotonic dynamics of influence functions throughout the learning process.

## 2.2 FROM STATIC TO DEVELOPMENTAL INFLUENCE FUNCTIONS

The stagewise development of singular models requires a corresponding shift in our tools for data attribution, moving from global, point-wise measures to local, distributional ones.

**Classical influence functions: a static view.** Classical influence functions (IFs) are a standard technique for training data attribution, quantifying how an infinitesimal upweighting of a training point $\mathbf{z}_i$ affects an observable $\phi$ evaluated at the final model parameters $\boldsymbol{w}^*$ (Cook, 1977). The influence is given by:

$$\mathrm{IF}(\mathbf{z}_i, \phi) = \frac{\partial}{\partial \beta_i} \phi(\boldsymbol{w}^*(\boldsymbol{\beta}))\Big|_{\boldsymbol{\beta}=\mathbf{1}} = -\nabla_{\boldsymbol{w}} \phi(\boldsymbol{w}^*)^\top \boldsymbol{H}^{-1}(\boldsymbol{w}^*) \nabla_{\boldsymbol{w}} \ell_i(\boldsymbol{w}^*), \tag{1}$$

where $\boldsymbol{H}(\boldsymbol{w}^*)$ is the Hessian of the total loss evaluated at the solution $\boldsymbol{w}^*$.

Crucially, the classical IF relies on the same regularity assumptions required for the Bernstein–von Mises theorem: the existence of a single, stable local minimum $\boldsymbol{w}^*$, and an invertible loss Hessian $\boldsymbol{H}(\boldsymbol{w}^*)$ at that minimum. As discussed, singular models like neural networks violate these conditions. Their loss landscapes are degenerate, featuring non-isolated minima and rank-deficient Hessians. This renders the classical IF theoretically ill-defined and practically unstable, which requires a dampening factor (see Appendix C.4), especially at intermediate checkpoints that are unconverged and away from minima.

**Bayesian influence functions: a developmental tool.** The Bayesian Influence Function (BIF) provides a principled alternative that is well-suited to the dynamics of singular models (Giordano et al., 2017; Kreer et al., 2025). Instead of measuring the change in a point estimate $\phi(\boldsymbol{w}^*)$, the BIF measures how the *posterior expectation* of an observable $\mathbb{E}[\phi(\boldsymbol{w})]$ changes. This derivative is equivalent to the negative covariance between the observable and the sample's loss:

$$\mathrm{BIF}(\mathbf{z}_i, \phi) = \frac{\partial}{\partial \beta_i} \mathbb{E}_{p_{\boldsymbol{\beta}}(\boldsymbol{w}|\mathcal{D})}[\phi(\boldsymbol{w})]\Big|_{\boldsymbol{\beta}=\mathbf{1}} = -\mathrm{Cov}_{p(\boldsymbol{w}|\mathcal{D})}(\ell_i(\boldsymbol{w}), \phi(\boldsymbol{w})). \tag{2}$$

This formulation is ideal for studying development for three main reasons:

- It is **distributional**. Its definition in terms of expectations over posteriors makes it a natural tool for the Bayesian learning framework of SLT.
- It is inherently **Hessian-free**. By replacing the problematic Hessian inverse with a covariance estimation, it remains well-defined even on degenerate loss landscapes.
- It is **well-defined at any point** in the training trajectory, not just at stable local minima. This is essential for studying influence as a dynamic quantity that evolves over time.

Moreover, when the regularity assumptions hold, the BIF asymptotically recovers the classical IF in the large-data limit (Kreer et al., 2025); that is, the BIF is a natural higher-order generalization of the classical influence function. For these reasons, we adopt the BIF as our primary tool for measuring influence.

**Estimating (local) Bayesian influence functions.** To measure the BIF in practice, we use an estimator based on stochastic-gradient MCMC introduced in Kreer et al. (2025). This also introduces a dampening term that enables localizing the BIF to individual model checkpoints. For more details, see Appendix B, where we also discuss practical scaling advantages of the BIF compared to other popular IF methods, such as EK-FAC (Grosse et al., 2023).

## 2.3 STAGEWISE DATA ATTRIBUTION

**Influence and susceptibility.** In the language of statistical physics, the BIF is an example of a generalized *susceptibility* that measures a system's response (in this case, a model's loss on sample $j$) to a perturbation (in this case, change in importance of sample $i$). In physical systems, susceptibilities diverge at phase transitions, which makes them macroscopically the most legible signal that a phase transition is taking place. This suggests using (Bayesian) influence functions to *discover* transitions during the learning process.

**Setup: a bimodal posterior.** A first-order phase transition is characterized by the posterior distribution $p(\boldsymbol{w} \mid \mathcal{D})$ having significant mass in two distinct neighborhoods, which we label $\mathcal{U}$ and $\mathcal{V}$. We can model this as a mixture distribution:

$$p(\boldsymbol{w} \mid \mathcal{D}) = \pi_{\mathcal{U}} p(\boldsymbol{w} \mid \mathcal{U}) + \pi_{\mathcal{V}} p(\boldsymbol{w} \mid \mathcal{V})$$

where $\pi_{\mathcal{U}}$ and $\pi_{\mathcal{V}}$ are the posterior probabilities of being in phase $\mathcal{U}$ or $\mathcal{V}$ respectively, with $\pi_{\mathcal{U}} + \pi_{\mathcal{V}} = 1$. At the peak of a phase transition, $\pi_{\mathcal{U}} \approx \pi_{\mathcal{V}} \approx 0.5$. Away from the transition, one of the weights is close to 1 and the other is close to 0.

**Decomposing influence with the law of total covariance.** The BIF between samples $i$ and $j$ is defined as $\mathrm{BIF}(\mathbf{z}_i, \ell_j) = -\mathrm{Cov}_{p(\boldsymbol{w}|\mathcal{D})}(\ell_i(\boldsymbol{w}), \ell_j(\boldsymbol{w}))$. We can decompose this total covariance using the Law of Total Covariance, conditioning on the phase ($Z \in \{\mathcal{U}, \mathcal{V}\}$):

$$\mathrm{Cov}(\ell_i, \ell_j) = \underbrace{\mathbb{E}[\mathrm{Cov}(\ell_i, \ell_j \mid Z)]}_{\text{Average Within-Phase Influence}} + \underbrace{\mathrm{Cov}(\mathbb{E}[\ell_i \mid Z], \mathbb{E}[\ell_j \mid Z])}_{\text{Between-Phase Influence}}$$

Let's analyze each term:

1. **Average Within-Phase Influence:** This term is the weighted average of the influence calculated strictly within each phase:

$$\mathbb{E}[\mathrm{Cov}(\ell_i, \ell_j \mid Z)] = \pi_{\mathcal{U}} \mathrm{Cov}_{\mathcal{U}}(\ell_i, \ell_j) + \pi_{\mathcal{V}} \mathrm{Cov}_{\mathcal{V}}(\ell_i, \ell_j)$$

   This represents the "baseline" influence. If there were no phase transition (e.g., $\pi_{\mathcal{U}} = 1$), this is the only term that would exist.

2. **Between-Phase Influence:** This term captures the covariance that arises because the expected losses themselves change as the model switches phases. Let $\mu_{i,\mathcal{U}} = \mathbb{E}[\ell_i \mid \mathcal{U}]$ be the expected loss of sample $i$ in phase $\mathcal{U}$, and likewise for $j$ and $\mathcal{V}$. The term expands to:

$$\mathrm{Cov}(\mathbb{E}[\ell_i \mid Z], \mathbb{E}[\ell_j \mid Z]) = \pi_{\mathcal{U}} \pi_{\mathcal{V}} (\mu_{i,\mathcal{U}} - \mu_{i,\mathcal{V}})(\mu_{j,\mathcal{U}} - \mu_{j,\mathcal{V}})$$

**Predicting stagewise changes in influence.** This decomposition predicts dynamic changing influence patterns over learning. The departure from the classical view arises because influence is phase-dependent: the baseline "within-phase" influence may differ significantly across the transition ($\mathrm{Cov}_{\mathcal{U}} \neq \mathrm{Cov}_{\mathcal{V}}$), and the "between-phase" term introduces an additional effect during the transition. In particular, we derive two predictions that diverge from the classical view:

- **Influence Can Change Sign:** If the within-phase influences of the two phases have significantly different values or if the between-phase term is large enough to dominate the average baseline influence during the transition, then transitions can cause a large change in magnitude or even a change in sign.

- **Inter-Phase Influence Peaks at Transitions:** The between-phase influence term is maximized when the posterior mass is evenly split ($\pi_{\mathcal{U}} \approx \pi_{\mathcal{V}} \approx 0.5$), which under some conditions may cause a peak in total influence at the critical point of a transition. The magnitude of the between-phase term is proportional to $(\mu_{i,\mathcal{U}} - \mu_{i,\mathcal{V}})$, which means the influence spike is largest for the samples on which the two phases disagree the most: peaks in influence identify the specific samples that *characterize* a given transition.

**Towards stagewise data attribution.** These theoretical predictions call for a shift from the classical static view of training data attribution to what we term *stagewise data attribution*: the analysis of influence as a dynamic trajectory over the entire learning process. The goal is to attribute learned behaviors not only to the data that influenced it, but also to the specific period of time in which that data had its effect. In the rest of this paper, we turn to testing the basic predictions above in order to validate this developmental framework.

## 3  TOY MODEL OF DEVELOPMENT

The acquisition of semantic knowledge involves the dynamic development of hierarchical structure in neural representations. Often, broader categorical distinctions are learned prior to finer-grained distinctions, and abrupt conceptual reorganization marks the non-static nature of knowledge acquisition, studied in both psychology and deep learning literature (Keil, 1979; Inhelder & Piaget, 1958; Hinton, 1986; Rumelhart & Todd, 1993; McClelland, 1995; Rogers & McClelland, 2004). For a principled understanding of the dynamical aspect of influence over training, we study a toy model of hierarchical feature learning introduced by Saxe et al. (2019a), for which ground truth and analytical tools are accessible.

We confirm our theoretical predictions in Section 2.3 by finding that the dynamics of influence coincide with the stagewise development of hierarchical structure. Sign flips occur as the model shifts to learning progressively finer levels of hierarchical distinctions, and peaks occur when the model begins learning a new level of distinction.

**Setup: a hierarchical semantic dataset.** We train a 2-layer deep linear network with MSE loss on a hierarchical semantic dataset from Saxe et al. (2019a). The dataset consists of one-hot input vectors representing objects, and each input maps to an output vector representing a collection of features that the object possesses (Figure 2). Importantly, this toy model mathematically shows that deep neural network architecture develops neural representation that reflects hierarchical differentiation in a progressive manner: learning animal vs. plants first, then mammals vs. birds, and then dogs vs. cats. See Appendix C for a detailed description of the toy model and its analytical treatment.

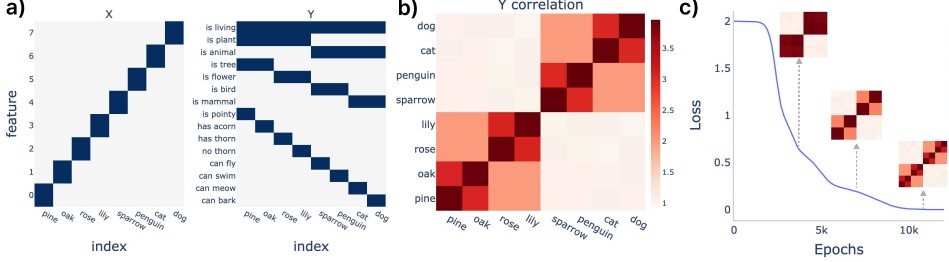

Figure 2: **A toy model of hierarchical semantic knowledge acquisition.** (a) A toy dataset adopted from Saxe et al. (2019a). Each object maps to a feature vector that describes the hierarchical structure of semantic knowledge, e.g., 'penguin' and `cat` are all living animals, but 'penguin' is a bird and `cat` is a mammal. (b) The correlation matrix of the feature output shows a hierarchical structure. (c) The hierarchical structure is acquired progressively during the training of the deep linear neural network.

**Measuring influence *dynamics*.** First, we probe the local BIF (Kreer et al., 2025) on the toy model over the entire learning trajectory. We use RMSProp-preconditioned SGLD sampler (described in Appendix B) to estimate a posterior from each checkpoint $\boldsymbol{w}_t^*$ at training time $t$. See Appendix B for details of the local BIF implementation and Appendix C.2 for the hyperparameter sweep. Furthermore, we derive the dynamics of the influence function analytically, leveraging the mathematical tractability of the toy model (see Appendix C.6).

**Leave-one-out (LOO) verification.** To confirm our observation from the BIF and analytical treatment, we conduct retraining experiments. Specifically, we consider the Leave-One-Out (LOO)

setting, where we ablate one data point and measure the loss difference of other data points compared to the baseline loss without ablation. We measure the loss difference *over training time* $t$:

$$\Delta\ell_{j,t}^{\backslash i} = \ell_{j,t}^{\mathcal{D}} - \ell_{j,t}^{\mathcal{D}_{\backslash i}}, \tag{3}$$

measured at time $t$ where $\mathcal{D}$ is the full dataset, $\mathcal{D}_{\backslash i}$ is the dataset with data index $i$ ablated, and $j$ is the index of the data point that we are querying. In Figure 3, we observe that the loss difference from LOO results in a similar pattern to what we see in the BIF and the analytical derivation, and validates their use. The additional results on perturbing other data points show a consistent trend in Appendix C.2 and are also visible in the dampened classical IF Appendix C.4.

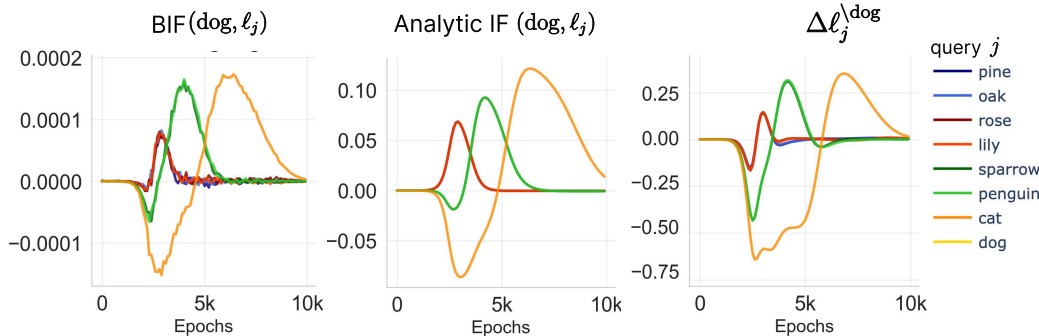

Figure 3: **Influence over time on a hierarchical semantic dataset.** We measure the influence of `dog` sample on other query samples $j$ with the following: (Left) BIF ($\beta = 1000, \epsilon = 1e\text{-}3, \gamma = 5e + 3$). (Center) Analytical IF (see full derivation in Appendix C.6). (Right) Loss difference from Leave-One-Out (LOO) retraining experiment. All three measures agree that the influence one sample has on another can vary non-monotonically over the course of training as discussed in Section 3. See Appendix C.2 for additional pairs of samples and experimental details.

With these results, we confirm the predictions from Section 2: (1) influence changes over time non-monotonically and can change sign, and (2) peaks in influence are correlated with key developmental transitions in model behavior. In the rest of the section, we will establish how these observations are reflected in the progressive learning of the hierarchical structure in our toy model.

**"When" matters for measuring influence.** We clearly observe that influence goes through non-monotonic change over the course of training, strengthening our argument that a static interpretation of influence is fallacious. In Appendix C.6, we analytically derive that influence is a function of singular mode strength of data input-output covariance learned by the network, which is a time-dependent variable and thus justifies studying data attribution from a dynamic perspective. Over the course of training, each data point induces a non-static influence on other data points. The influence can flip sign—the same data can be either helpful or harmful, depending on when it is presented. It can also mark clear peaks at a specific time point on different query data points. Furthermore, the influence over time from one data point to another point is specific to those points. The influence from `dog` to `cat` might be the same as `dog` to 'penguin' early on, but they are distinguished later in the learning process.

To confirm that these influence dynamics represent a causal effect on model performance, we perform an additional retraining experiment in Appendix C.5. Instead of removing a sample for the full training run, we ablate specific data points only during short temporal windows. We find that ablating data precisely during the stage where the BIF assigns its peak influence induces the largest loss difference compared to ablation at other times. This demonstrates that our measure correctly identifies the critical window in which a specific sample drives the learning process.

**Change of influence reflects stagewise learning.** During the phase where one hierarchical distinction (e.g., `animal` vs. `plant`) is being learned, upweighting a data point in the same class (`dog`) is helpful for learning a query data point (`sparrow`) as indicated by a negative influence (positive covariance). In contrast, upweighting a data point that belongs to a different class harms learning that query data point (`pine`), reflected in a positive influence. A data point `dog` is helpful (negative

influence) to learning `sparrow` early on in the learning, while learning to distinguish `animal` vs. `plant`, but it is harmful (positive) later on when learning to distinguish `mammal` vs. `bird`. In Figure 4 b), we show Multi-Dimensional Scaling (MDS, Torgerson 1952; Cox & Cox 2008) of the hidden representation of each data over time as in Saxe et al. (2019a) where learning of each hierarchical level is reflected on the branching node (numbered). We observe that the time point of the branching node matches the peaks in influence. That is, at the transition, where the model learns to distinguish `mammal` vs `bird` within the animals and form a new hierarchy level, the influence between `mammal` and `bird` is the highest.

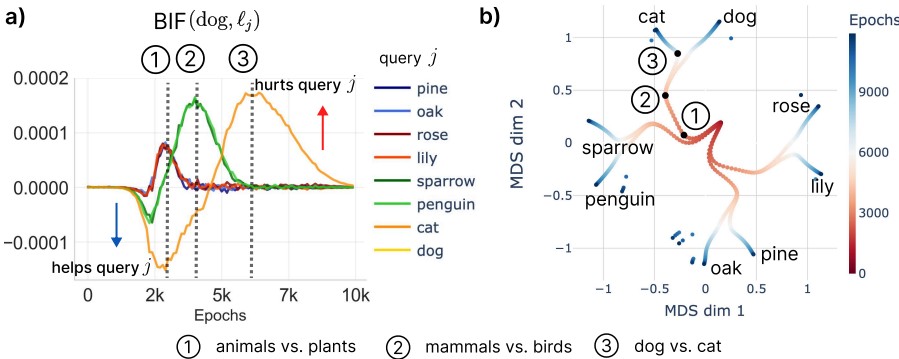

Figure 4: **BIF captures developmental influence.** (a) The peak positive influence from `dog` to different data points is noted with ①, ②, and ③. (b) MDS of the hidden representations in the network over the course of learning. The peaks in influence match the branching points of the MDS trajectory, where each hierarchical category develops (black points).

## 4 LANGUAGE MODELS

To investigate the dynamics of influence in a real-world setting, we study the acquisition of token-level syntactic knowledge using language models from the Pythia Scaling Suite (Biderman et al., 2023).

We confirm our theoretical predictions in Section 2.3 by finding non-monotonic influence trajectories, with large changes in magnitude, sign flips, and peaks that correspond to known developmental changes like the formation of the induction circuit.

**Per-token influence functions.** One clear benefit of the BIF in the language-modeling setting is that computing influence at the level of individual tokens rather than sequences incurs no additional computational cost. Loss is computed on a per-token basis during RMSprop-SGLD, which is already necessary when considering the autoregressive losses that represent the standard for LLM pretraining:

$$\ell_i(\boldsymbol{w}) = \sum_k \ell(x_{i,k} \mid x_{i,0} \ldots x_{i,k-1}, \boldsymbol{w}),$$

where $x_{i,j}$ is the $j$th token in the $i$th text sequence. These per-token losses can then be stored individually and used to estimate the per-token BIF matrix for the relevant dataset.

**Classifying tokens into syntactic classes.** Following the experimental setup of Baker et al. (2025), we classify individual tokens according to how they are used to give structure to text. These classes include strictly syntactic tokens (left delimiters, right delimiters, and formatting tokens—such as newlines), morphological roles (parts of words and word endings), and a broader structural class—tokens that have been used in conjunction earlier in the context, forming an inductive pattern. We note that this classification is not exhaustive, nor is it exclusive—not all tokens have a class, and some may occupy multiple classes at once. A full classification is provided in Appendix D.1.1.

**Calculating group influence.** To estimate patterns of influence between tokens across classes, we make use of the following procedure:

1. Using a subset of The Pile (Gao et al., 2021), we compute the *normalized* BIF (Kreer et al., 2025) between all pairs of tokens, sampled from the SGLD-estimated model posterior, as described in Appendix B.

2. Each token is classified in accordance with the listed structural classes based on pattern-matching, following Baker et al. (2025).

3. For every possible pair of classes, we compute the average (or "group") influence between tokens in these classes, excluding influences between the same token in different classes.

These inter-class influences can then be used to provide insight into how much influence tokens from one class have on tokens from other classes.

**Learning induction.** In Figure 5, we plot the inter-class influence for each query class across all class pairings at several model checkpoints. From these plots, some clear dynamics emerge. Using the BIF, we see that the formation of model structure corresponding to induction relationships starts as early as 128 steps into learning, when there is an inflection point in the average BIF between induction tokens. This continues to strengthen for the next 30k training steps before appearing to peak and fall. This echoes the results of Tigges et al. (2024), which finds that Pythia models begin to learn the induction circuit at this time, and exhibit an apex at 30k training steps before diminishing.

In Appendix D.3, we study a small language model trained from scratch and track the influence on a targeted set of synthetic samples to automatically identify the induction bump. We find that the influence of induction-pattern tokens sharply rises as the induction circuit is being learned. We further validate the practical utility of these insights with a stagewise intervention experiment in Appendix D.4. We find that upweighting induction patterns specifically when the induction circuit begins to form accelerates the formation of induction heads significantly more than upweighting them before this window. This confirms the prediction derived from the influence dynamics: influence cannot be fully understood from a static analysis at the end of training.

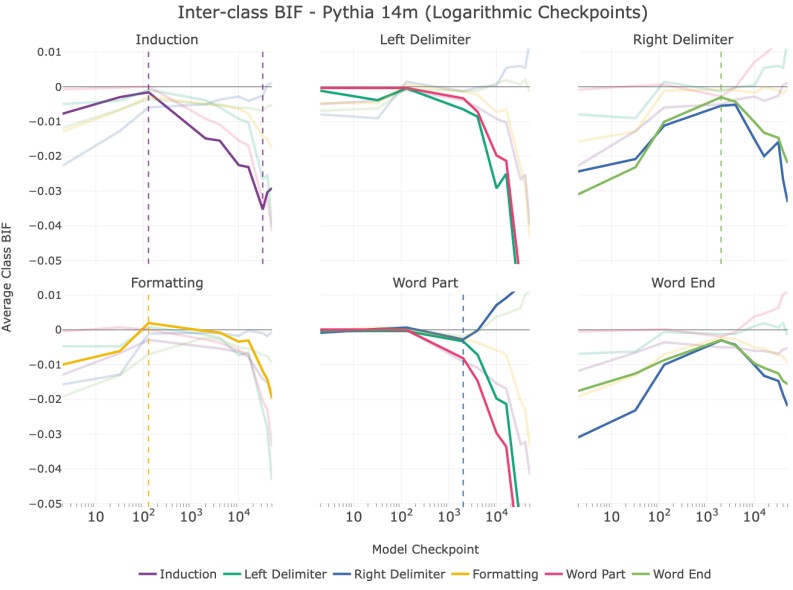

Figure 5: **Token–class relationships.** BIF between structural classes through training. We observe structural relationships between tokens reflected in influence patterns between classes, including strong intra-class relationships, development of induction, and relationships between word elements and corresponding delimiter token classes. Dashed lines indicate major inflection points in the BIF. Opacity captures notable influence patterns.

**Learning where to end.** Another notable pattern occurs in the influence traces of left delimiters and right delimiters. Both classes appear among constructive (negative) influence tokens for the

other class very early in training, but this relationship quickly inverts, with tokens from the opposite delimiter class becoming increasingly positive as the model learns to distinguish between these classes. We also see increasing influence between left and right delimiters and related classes that may perform similar roles in structuring text. Left delimiters share a similar influence pattern overall with parts of words but not the ends of words. while right delimiters influence ends of words but not word parts. These relationships are also visible in the reverse direction (from word parts/word ends to left/right delimiters). These relationships develop over time, with a particularly notable spike in the relationship between word parts and left delimiters later in model training.

**Dynamics of influence.**    Taken together, these results demonstrate that influence between different classes of tokens is not a static property but a dynamic one that evolves throughout training. We observe changes in both the magnitude and sign of influence, with the timing of these shifts varying depending on the specific structural capability being learned. For instance, the influence related to induction patterns exhibits a non-monotonic peak that aligns with the known developmental phase transition for this circuit. Similarly, the relationship between left and right delimiters, as well as the relationship between word parts/word ends and left/right delimiters undergoes a sign flip, indicating a qualitative shift in how the model processes scope and pairing. These findings confirm the predictions of our stagewise framework: different data becomes influential at different times, reflecting the model's progressive, stage-by-stage acquisition of syntactic and structural knowledge.

## 5    Discussion & Conclusion

**Stagewise data attribution and developmental interpretability.**    This work places data attribution within the context of Developmental Interpretability. We follow the established methodology of using singular learning theory (SLT) as a theoretical lens to study the dynamics of models trained with stochastic optimizers. Prior work has successfully used this pipeline to explain phenomena like phase transitions in toy models of superposition (Chen et al., 2023), algorithm selection in transformers (Carroll et al., 2025), stagewise learning in toy transformers (Urdshals & Urdshals, 2025), and the stagewise emergence of structure (e.g., n-grams, induction, parenthesis-matching, space-counting) in language models (Hoogland et al., 2024; Wang et al., 2025b; Baker et al., 2025; Wang et al., 2025a). Our contribution is to use the SLT account of phase transitions to predict and subsequently verify that a training sample's influence is not static but a dynamic quantity that evolves with the model's development.

**Stagewise vs. unrolling-based attribution.**    Our work complements another important line of research that moves beyond static influence functions: trajectory-based or "unrolling" methods like TracIn (Pruthi et al., 2020), HyDRA (Chen et al., 2021), and SOURCE (Bae et al., 2024). These techniques approximate the total influence of a sample by integrating its contributions (such as gradient updates or local influence scores) across numerous checkpoints along the full training path. This approach provides a more faithful account of the path-dependent nature of SGD and can offer more accurate attribution scores than single-point estimates.

However, our findings regarding influence dynamics raise the possibility of cancellation effects in these unrolling-based measures. Since influence can flip signs during training, opposing contributions from different stages could potentially offset each other when integrated. While such a score may still accurately reflect the expected cumulative impact, it risks obscuring the data point's role in driving specific developmental stages. Fundamentally, the goal of these methods differs from ours: unrolling techniques compute a single, cumulative summary of a sample's total impact, treating the learning process as a black box. In contrast, our framework treats the learning trajectory as an object of study in itself, aiming to understand *when* data matters during development.

**A mechanism for implicit curricula.**    The concept of a curriculum—learning from easier to harder data—has been argued consistently for more efficient neural learning (Bengio et al., 2009; Wang et al., 2021; Lee et al., 2024). However, the efficacy of this technique has been shown to be limited in practice (Wu et al., 2020; Mannelli et al., 2024). One prevalent explanation is that an *implicit curriculum* is adopted by learning through gradient descent in neural networks (Graves et al., 2017; Rahaman et al., 2019; Saxe et al., 2019a; Valle-Perez et al., 2018). Our findings provide a new, more granular mechanism for understanding this phenomenon. The stagewise evolution of influence

demonstrates how different data points become "important" at different moments, effectively creating a dynamic, self-organizing curriculum.

**Limitations and future work.** Our framework points toward several avenues for future research. The primary theoretical gap remains the link between the Bayesian learning process of SLT and the non-equilibrium dynamics of SGD. On the empirical front, a key direction is to move from a behavioral to a mechanistic account of influence. Ultimately, model generalization is grounded in the circuits and internal structure a model acquires over training. A more complete science of interpretability therefore understands not just *which samples* are most influential (data attribution), but *when* they are most influential (stagewise attribution), and *how* they shape model internals such as features and circuits (mechanistic interpretability).

**From pointwise to stagewise data attribution.** Ultimately, this paper argues for a shift in how we approach training data attribution. The static perspective, which assigns a single, global influence score to each data point, only at the end of training, offers an incomplete and, at times, misleading picture. By demonstrating that influence is inseparable from development, we advocate for moving from point-wise to stagewise data attribution.

This developmental lens is essential for tackling a key scientific challenge: understanding the correspondence between data structure, loss landscape geometry, learning dynamics, and model internals (Wang et al., 2025b). Stagewise data attribution, by tracking influence *dynamics*, provides a concrete tool to map these connections, opening new possibilities for interpreting, debugging, and ultimately steering how models learn.

## ACKNOWLEDGMENTS

We would like to thank Simon Pepin Lehalleur for his detailed feedback on an earlier version of this manuscript. We are also grateful to Daniel Murfet, Andrew Saxe, Andrew Lampinen, Aaditya Singh, and Philipp Alexander Kreer for their valuable feedback and insightful discussions. We thank Rohan Hitchcock for his helpful input on hyperparameter calibration.

Jin Hwa Lee and Matthew Smith gratefully acknowledge the support of the Pivotal Mentorship Program, and thank Morgan Simpson and Euan McLean for their guidance, as well as Guillaume Corlouer and Avi Semler for their insights. Jin Hwa Lee thanks the Gatsby Charitable Foundation (GAT3755) and The Wellcome Trust (219627/Z/19/Z). Jin Hwa Lee and Matthew Smith would also like to thank LISA for providing a productive working environment and support for this project. Finally, we thank Stan van Wingerden for his assistance with the compute infrastructure.

## AUTHOR CONTRIBUTIONS

Jin Hwa Lee developed, implemented, and analyzed the toy model experiments. Matthew Smith designed and conducted the language model experiments. Maxwell Adam helped set up the language modeling experiments and contributed the induction-formation investigation. Jesse Hoogland conceived the project, developed the theoretical framework, and led the research. All authors contributed to the writing of the manuscript.

## REPRODUCIBILITY STATEMENT

We describe our primary method, the Bayesian Influence Function (BIF) estimated via RMSPropS-GLD, in Appendix B. We first validate our predictions in a controlled toy model (Section 3), providing analytical derivations, Leave-One-Out (LOO) verification, and all hyperparameters in Appendix C. We then demonstrate our findings at scale using public Pythia language models on The Pile dataset (Section 4), with all experimental specifics, token classification schemes, and hyperparameters detailed in Appendix D.

## LLM USAGE STATEMENT

We used large language models (LLMs) to assist in the preparation of this manuscript. Their role included proofreading, correcting grammatical errors, and rephrasing sentences to improve clarity and flow. Beyond surface-level edits, we utilized LLMs as a collaborative tool for brainstorming options for narrative structures and receiving feedback on the clarity of our arguments. LLMs also served as coding assistants in implementing our experiments and generating visualizations. Additionally, we used LLMs to aid in the research process, for instance, by suggesting relevant theoretical concepts such as the Law of Total Covariance for the decomposition in Section 2. In all instances, the final content, including all code, theoretical claims, and text, was reviewed, validated, and is the sole responsibility of the authors.

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

# Appendix

1. **Appendix A: Phase Transitions.** This section provides a formal outline of how phase transitions arise in singular models and how they impact influence functions.

2. **Appendix B: Bayesian Influence Functions.** This section provides additional experimental details on estimating the local BIF.

3. **Appendix C: Toy Model Details and Additional Results.** This subsection describes the hierarchical dataset, model architecture, and Leave-One-Out (LOO) training procedure. It also contains additional LOO results and the full analytical derivation of the influence dynamics in the deep linear network model.

4. **Appendix D: Language Model Details and Additional Results.** This subsection describes the structural token classification scheme, provides RMSPropSGLD hyperparameters for the Pythia experiments, and details additional validation experiments.

## A   PHASE TRANSITIONS

In this appendix, we provide a theoretical justification for the claims made in Section 2 regarding phase transitions and their effect on model predictions and influence, grounded in statistical physics and the theory of Bayesian statistics. This theoretical treatment motivates our empirical investigation into the interaction between stagewise development and influence functions for models trained via standard stochastic optimizers.

### A.1   BIF AS SUSCEPTIBILITY

**Setup.**   We begin with the model-truth-prior triplet:

1. The **model** $p(\mathbf{z} \mid \boldsymbol{w})$ assigns a probability to a sample $\mathbf{z} \in \mathcal{Z}$ for a given choice of weights $\boldsymbol{w} \in \mathcal{W} \subseteq \mathbb{R}^d$.

2. The **truth** or data-generating distribution $q(\mathbf{z})$ generates a dataset of i.i.d. samples $= \{\mathbf{z}_i\}_{i=1}^n$.

3. The **prior** $\varphi(\boldsymbol{w})$ assigns an initial probability distribution to each choice of weights.

From this triplet, we obtain a posterior through (the repeated application of) Bayes' rules:

$$p(\boldsymbol{w} \mid \mathcal{D}) = \frac{p(\mathcal{D} \mid \boldsymbol{w})\varphi(\boldsymbol{w})}{p(\mathcal{D})}, \tag{4}$$

where $p(\mathcal{D} \mid \boldsymbol{w}) = \prod_i p(\mathbf{z}_i \mid \boldsymbol{w})$ is the likelihood, and $p(\mathcal{D}) = \int_{\mathcal{W}} p(\mathcal{D} \mid \boldsymbol{w})\varphi(\boldsymbol{w})d\boldsymbol{w}$ is the marginal likelihood. Often, we're interested in a *tempered* Bayesian posterior:

$$p_{\boldsymbol{\beta}}(\boldsymbol{w} \mid \mathcal{D}) = \frac{p_{\boldsymbol{\beta}}(\mathcal{D} \mid \boldsymbol{w})\varphi(\boldsymbol{w})}{p_{\boldsymbol{\beta}}(\mathcal{D})} = \frac{e^{-\boldsymbol{\beta}\cdot\boldsymbol{\ell}(\boldsymbol{w})}\varphi(\boldsymbol{w})}{p_{\boldsymbol{\beta}}(\mathcal{D})}, \tag{5}$$

where $\boldsymbol{\beta}$ is a vector of per-sample importance weights, and $\boldsymbol{\ell}(\boldsymbol{w})$ is the vector of per-sample losses for a given weight $\boldsymbol{w}$.

**The (global) free energy formula.**   The (global) free energy is the negative log marginal likelihood, $F_n^{\boldsymbol{\beta}} = -\log p_{\boldsymbol{\beta}}(\mathcal{D})$. A central result of singular learning theory (SLT; Watanabe 2009) is an asymptotic (in the limit of infinite data) expression for this quantity. For a critical point $\boldsymbol{w}^*$ in a neighborhood $\mathcal{W}$, the free energy asymptotically expands as

$$F_n := F_n^{\boldsymbol{\beta}}\bigg|_{\boldsymbol{\beta}=\mathbf{1}} = nL_n(\boldsymbol{w}^*) + \lambda(\boldsymbol{w}^*)\log n + O_p(\log\log n), \tag{6}$$

where $\lambda(\boldsymbol{w}^*)$ is the learning coefficient, a degeneracy-aware measure of complexity (that coincides with half the effective parameter count for minimally singular models).

**Free energy as moment-generating function.** In statistical physics and thermodynamics, the free energy is the central object of study. This is because, if one manages to obtain a closed-form expression for the free energy, it is possible to calculate arbitrary expectation values simply by differentiating the free energy. Expectation values are of primary interest because they correspond to the things we can measure in an experimental setting.

For example, in the learning theory setting, the expected (per-sample) loss can be expressed as

$$\mathbb{E}[\ell_i(\boldsymbol{w})] = \left.\frac{\partial F_n^{\boldsymbol{\beta}}}{\partial \beta_i}\right|_{\boldsymbol{\beta}=\mathbf{1}}. \tag{7}$$

Combining Equation (7) with Equation (2), lets us express the BIF as a 2nd-order derivative of the free energy, assuming both samples are in the training dataset:

$$\mathrm{BIF}(\mathbf{z}_i, \mathbf{z}_j) = \left.\frac{\partial \mathbb{E}_{\boldsymbol{\beta}}[\ell_j(\boldsymbol{w})]}{\partial \beta_i}\right|_{\boldsymbol{\beta}=\mathbf{1}} = \left.\frac{\partial^2 F_n^{\boldsymbol{\beta}}}{\partial \beta_i \partial \beta_j}\right|_{\boldsymbol{\beta}=\mathbf{1}}. \tag{8}$$

Susceptibilities are defined as second-order derivatives of the free energy, which makes the BIF an example of a generalized susceptibility.

## A.2 INTERNAL MODEL SELECTION

**The (local) free energy formula.** The local free energy formula is defined analogously to the global free energy, but with the domain of integration restricted to a particular region of parameter space or "phase" $W \subseteq \mathcal{W}$:

$$F_n^{\boldsymbol{\beta}}(W) = -\log \int_W p_{\boldsymbol{\beta}}(\mathcal{D} \mid \boldsymbol{w})\varphi(\boldsymbol{w})d\boldsymbol{w}. \tag{9}$$

This admits an analogous asymptotic form to Equation (6):

$$F_n(W) := \left.F_n^{\boldsymbol{\beta}}(W)\right|_{\boldsymbol{\beta}=\mathbf{1}} = nL_n(\boldsymbol{w}^*) + \lambda(\boldsymbol{w}^*)\log n + O_p(\log\log n), \tag{10}$$

but where now $\boldsymbol{w}^*$ is a local minimum within $W$, and $\lambda$ is the local learning coefficient associated with that local minimum (Lau et al., 2025).

**Coarse-graining.** Given a partitioning of parameter space into multiple disjoint "phases" $\mathcal{W} = \cup_i W_i$, the global free energy can be computed from the local per-phase free energies as follows:

$$F_n = -\log \int_{\mathcal{W}} e^{-nL_n(\mathcal{D}|\boldsymbol{w})}\varphi(\boldsymbol{w})d\boldsymbol{w} \tag{11}$$

$$= -\log \sum_i \int_{W_i} e^{-nL_n(\mathcal{D}|\boldsymbol{w})}\varphi(\boldsymbol{w})d\boldsymbol{w} \tag{12}$$

$$= -\log \sum_i e^{-F_n(W_i)} \tag{13}$$

$$\approx \min_i F_n(W_i). \tag{14}$$

The last line follows from the well-known use of a log-sum exponential as a smooth approximation for the max function. This is to say that globally, the free energy is determined primarily by the phase with the lowest free energy, with exponentially suppressed contributions from all other regions of phase space.

**Competition between phases.** Assume the posterior distribution is concentrated in just two distinct neighborhoods, which we label $\mathcal{U}$ and $\mathcal{V}$, and is vanishing everywhere else. That is, these two phases

constitute a degenerate set of minimizers of the free energy. We can then write the posterior as a mixture distribution:

$$p(\boldsymbol{w} \mid \mathcal{D}) = p(\boldsymbol{w} \mid \mathcal{U})p(\mathcal{U} \mid \mathcal{D}) + p(\boldsymbol{w} \mid \mathcal{V})p(\mathcal{V} \mid \mathcal{D})$$

where $p(\boldsymbol{w} \mid \mathcal{U})$ is the posterior distribution conditioned on the model being in phase $\mathcal{U}$, and $p(\mathcal{U} \mid \mathcal{D})$ is the total posterior probability of this phase, with $\pi_{\mathcal{U}} + \pi_{\mathcal{V}} \approx 1$.

**First-order phase transitions.** This free energy formula predicts the existence of first-order "Bayesian" phase transitions. When two solutions compete, e.g., $u$ with neighborhood $\mathcal{U}$ and $v$ with neighborhood $\mathcal{V}$, then the posterior log-odds evolve as $\log \frac{p_n(\mathcal{U})}{p_n(\mathcal{V})} = n\Delta L_n + \log(n)\Delta\lambda + O_p(\log\log n)$, where $\Delta L_n = L_n(v) - L_n(u)$ and $\Delta\lambda = \lambda(v) - \lambda(u)$. If $u$ has higher loss but lower complexity ($\Delta L_n < 0, \Delta\lambda > 0$), the posterior initially prefers the simple solution $\mathcal{U}$ but switches to the complex solution $\mathcal{V}$ when $\log(n)/n < -\Delta L_n/\Delta\lambda$. This is a "Type-A" transition (Carroll et al., 2025; Hoogland et al., 2025), see Figure 1.

Alternatively, if the two phases agree on the linear term (they have the same minimum loss), then the trade-off will be pushed down into lower-order terms, and there can be a "Type-B" transition, in which complexity (as measured by the LLC) *decreases*, in exchange for an increase in lower-order terms.

## B  BAYESIAN INFLUENCE FUNCTIONS

**Estimating the BIF with SGMCMC.** To estimate the BIF in practice, we use an RMSProp-preconditioned SGLD sampler (Welling & Teh, 2011; Li et al., 2016) to estimate a posterior from each checkpoint $\boldsymbol{w}^*$ This approximates Langevin dynamics with loss gradients and locality regularization from initial $\boldsymbol{w}^*$ modulated by localization strength $\gamma$,

$$\boldsymbol{w}_{s+1} = \boldsymbol{w}_s - \frac{\hat{\epsilon}_s}{2}\left(\sum_{i \in |\mathcal{D}|} \nabla_{\boldsymbol{w}}\ell_i(\boldsymbol{w}_s) + \gamma(\boldsymbol{w}_s - \boldsymbol{w}^*)\right) + \mathcal{N}(0, \hat{\epsilon}_s). \tag{15}$$

The update rule takes advantage of an adaptive learning rate $\hat{\epsilon}_s$ compared to vanilla SGLD and is more robust to varying step size (Hitchcock et al., 2025). The full algorithm is described in Algorithm 1.

**Practical considerations with the BIF.** As mentioned in the main body and following the discussion at length in Kreer et al. (2025), we consider several practical modifications of the local BIF. First, we use a normalized BIF for the language-modeling experiments, which involves computing the Pearson correlation instead of the covariance over losses. Empirically, we find that this behaves more stably than the raw covariance and thus is easier to track over time. Second, we consider a per-token BIF, which can be trivially obtained by avoiding the loss accumulation over token indices and saving all per-token losses at each SGLD draw. Finally, to avoid potentially spuriously high covariances, we drop same-token influence scores when computing aggregate group influence scores, as in Adam et al. (2025).

**Addressing influence from unseen training samples.** A potential objection to our developmental analysis is that at early checkpoints, the model's optimizer has not yet encountered every training sample. It might therefore seem paradoxical to measure the "influence" of a sample the model has not yet "seen."

This concern can be straightforwardly resolved. The BIF is defined as the sensitivity of an observable to a sample's contribution weight, $\beta_i$, to the total loss. For a sample $\mathbf{z}_i$ that has already been processed, we measure influence by considering perturbations around its baseline weight of $\beta_i = 1$. For a sample the optimizer has not yet encountered, we can simply treat its current weight as $\beta_i = 0$ and evaluate the same derivative at that point.

The resulting quantity remains well-defined, with a slightly different interpretation: it measures the model's sensitivity to the *initial introduction* of a new sample, rather than the *re-weighting* of an existing one. Our practical implementation naturally accommodates this, as we use independent data sources for the SGLD gradient updates and the forward passes used to compute the loss covariance.

---

**Algorithm 1** RMSPropSGLD for Bayesian influence

---

**Input:** Initial model parameters $\boldsymbol{w}^* \in \boldsymbol{w}$, training dataset $\mathcal{D} = (\boldsymbol{z}_i)_{i=1}^{n=8}$, loss functions $\ell_i :=$ $\ell(\boldsymbol{z}_i; -)\colon \boldsymbol{w} \to \mathbb{R}$ for each $i \in [n]$, observables $\phi_j \colon \boldsymbol{w} \to \mathbb{R}$ for each $j \in [n]$, RMSPropSGLD hyperparameters $\epsilon$ (step size), $\beta$ (inverse temperature), $\gamma$ (localization), $m$ (sampling batch size), $C$ (number of chains), $T$ (chain length), $b$ (decay rate), $\alpha$ (stability constant).
**Output:** $\boldsymbol{B} = (\mathrm{BIF}(\boldsymbol{z}_i, \phi_j))_{1 \leq i \leq n, 1 \leq j \leq m} \in \mathbb{R}^{n \times m}$
$\boldsymbol{L} \leftarrow \boldsymbol{0}_{n \times CT}, \boldsymbol{\Phi} \leftarrow \boldsymbol{0}_{m \times CT}$
**for** $1 \leq c \leq C$ **do**
    $\boldsymbol{w} \leftarrow \boldsymbol{w}^*$
    **for** $1 \leq t \leq T$ **do**
        **for** $1 \leq i \leq n$ **do**
            $\boldsymbol{L}_{i,(c-1)C+t} \leftarrow \ell_i(\boldsymbol{w})$            ▷ Compute train losses
        **end for**
        **for** $1 \leq j \leq m$ **do**
            $\boldsymbol{\Phi}_{j,(c-1)C+t} \leftarrow \phi_j(\boldsymbol{w})$            ▷ Compute observables
        **end for**
        Sample full batch $\mathcal{B} = \mathcal{D}$ of size $m = n$ for small toy dataset.
        $\boldsymbol{V}_t \leftarrow b\boldsymbol{V}_{t-1}[i] + (1-b)\nabla_w \ell^2$
        $\hat{\boldsymbol{V}}_t \leftarrow \frac{1}{1-b^t}\boldsymbol{V}_t$
        $\hat{\epsilon}_t \leftarrow \frac{\epsilon}{\sqrt{\hat{V}_t + \alpha}}$
        $\eta_t \leftarrow \eta \sim \mathcal{N}(0,1)$
        $\boldsymbol{w} \leftarrow \boldsymbol{w} - \frac{\hat{\epsilon}_t}{2}\left(\frac{\beta n}{m}\sum_{k \in \mathcal{B}_t}\nabla_{\boldsymbol{w}}\ell_k(\boldsymbol{w}) + \gamma(\boldsymbol{w} - \boldsymbol{w}^*)\right) + \sqrt{\hat{\epsilon}_t}\eta_t$     ▷ SGLD update
    **end for**
**end for**
$\boldsymbol{B} \leftarrow \frac{1}{CT-1}\boldsymbol{L}\left(\boldsymbol{I}_{CT} - \frac{1}{CT}\boldsymbol{1}_{CT}\boldsymbol{1}_{CT}^\top\right)^2 \boldsymbol{\Phi}^\top$     ▷ Covariance between $\boldsymbol{L}$ and $\boldsymbol{\Phi}$
**Return** $\boldsymbol{B}$

---

**Scalability of BIF.** Besides the theoretical advantages discussed in Section 2.2, the BIF also has several practical advantages over classical IF approximations, as discussed in Kreer et al. (2025).

First, classical approximations like EK-FAC (Grosse et al., 2023) incur a substantial up-front computational cost to "fit" the inverse Hessian estimate. The BIF, in contrast, has no upfront fitting cost. This comes at the cost of having a higher compute cost per individual query. This means that the BIF is more computationally efficient for smaller datasets, while the classical IF is more efficient for larger datasets, where it can amortize the upfront costs over many queries. For the settings we considered, where the focus is on studying development, the priority was coverage over many checkpoints rather than coverage over the entire dataset. These tradeoffs favored the BIF as the more tractable choice.

Furthermore, our focus on *fine-grained* structural attribution places us in the specific regime where BIF is most efficient: computing dense, per-token influence matrices for targeted subsets of data. Unlike Hessian-based methods, which typically require sequential scoring passes to isolate each individual token contribution, the BIF estimator computes the full set of token-by-token influences in a single batched process without additional memory overhead for backpropagating individual per-token gradients.

## C   TOY MODEL OF DEVELOPMENT

In this section, we provide details of the toy model experimental details and analytical investigation of deep linear neural network learning dynamics under the data perturbation introduced in Section 3. It includes a training setup of toy data with a 2-layer deep linear neural network and BIF hyperparameter sweeps, and extended results of the Leave-One-Out (LOO) experiments and BIF measurements.

## C.1   TOY DATASET

We use the semantic hierarchical dataset introduced in Saxe et al. (2019a). We set the number of data points $N = 8$ with hierarchy level $H = 3$, that is, the data can be organized by 4 levels of hierarchy, e.g., dog belongs to the lowest hierarchy level 1 of living organism, and level 2 of animals (vs. plants), and level 3 of mammals (vs. birds). The input is a data index given by an identity matrix of size $N \times N$. The model needs to learn to associate with the feature of each data point, which forms the above hierarchical structure, the output size of $N \times O$ with $O = 15$.

We train a 2-layer bias-free linear neural network with 50 hidden neurons, which is overparameterized for our rank 8 dataset. We initialize the network with small weights by sampling from a Gaussian distribution $\mathcal{N} \sim (0, \sigma^2)$, $\sigma = 1e^{-3}$. We train the network on mean squared error (MSE) loss with a small learning rate $5e^{-3}$ and train for $10K$ epochs with full batch unless mentioned otherwise.

## C.2   BIF ADDITIONAL RESULTS AND HYPERPARAMETERS SWEEP

For a principled decision of hyperparameters for the SGLD sampler, we perform a preliminary hyperparameter sweep on the SGLD sampler using the same deep linear neural network architecture and the data to estimate LLC. Since ground truth LLC is known for deep linear neural networks (Aoyagi, 2024), we can select a range of hyperparameters by choosing ones that have a well-matched LLC estimate to the ground truth. Based on this procedure, we validate the superiority of RMSPropSGLD to vanilla SGLD and also the insensitivity of the sampler quality to decay rate $b$ and stability constant $\alpha$ in a certain range and fixed the value to reduce the grid search dimension. We also narrowed down the localization strength range.

The range of the conducted hyperparameter sweeps for BIF measurement is summarized in Table 1. The values given in the range were sampled on a logarithmic scale. RMSPropSGLD sampling procedure is shown in B.

Table 1: Summary of hyperparameter sweep range for BIF experiments on toy model.

| Hyperparameter | Range |
| --- | --- |
| $\epsilon$ (step size) | [1e-7 - 1e-2] |
| $\beta$ (inverse temperature) | [1e+1 - 1e+4] |
| $\gamma$ (localization strength) | [1e-2 - 1e+6] |
| $m$ (batch size) | 8 (full batch) |
| $C$ (number of chains) | [2,4,8] |
| $T$ (chain length) | [200, 400, 800, 1000] |
| $b$ (decay rate) | [0.8, 0.9, 0.95, 0.99] |
| $\alpha$ (stability constant) | [1e-4 - 1.0] |

We present the Pearson correlation coefficient between Leave-One-Out (LOO) loss difference trajectory and BIF trajectory over learning over varying hyperparameter choices of inverse temperature $\beta$, localization strength $\gamma$ and step size $\epsilon$ in Figure 6. In Figure 6, we see that the correlation coefficient changes smoothly as we vary the hyperparameters. In the main  Section 3 Figure 3, we showed the BIF measured with the hyperparameters that have the highest correlation with LOO ($\beta = 1000, \epsilon = 1e\text{-}3, \gamma = 5e + 3$).

## C.3   LEAVE-ONE-OUT (LOO) EXPERIMENT

For the Leave-One-Out (LOO) experiment, which is also referred to as retraining, we mask the corresponding data index to 0 for both input and output. We use the same hyperparameter as above and we report the loss difference of data $j$ after ablating data $i$ at each time, $\Delta \ell_{j,t} = \ell_{j,t}^{\mathcal{D}} - \ell_{j,t}^{\mathcal{D} \setminus i}$.

## C.4   CLASSICAL INFLUENCE FUNCTION

We measure classical influence over the training trajectory of the toy model

$$\text{IF}_t(\mathbf{z}_i, \phi) = -\nabla_{\boldsymbol{w}_t} \phi(\boldsymbol{w}_t)^\top \boldsymbol{H}^{-1}(\boldsymbol{w}_t) \nabla_{\boldsymbol{w}_t} \ell_i(\boldsymbol{w}_t), \tag{16}$$

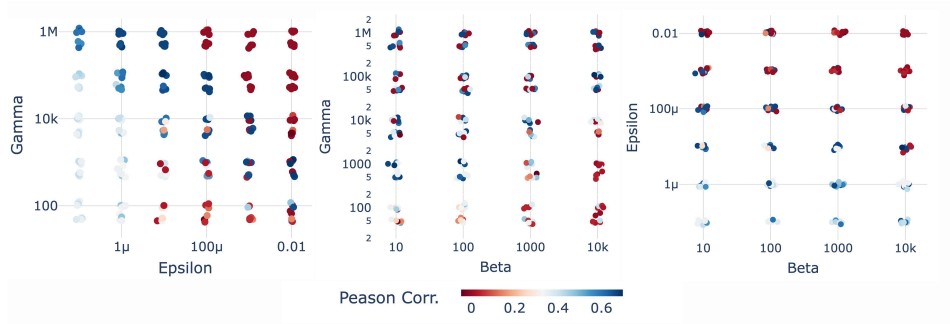

Figure 6: **BIF–LOO trace correlation with varying inverse temperature $\beta$, localization strength $\gamma$ and step size $\epsilon$.** On a single grid point between two hyperparameters (e.g. $\gamma$, $\epsilon$ in the leftmost), multiple points refer to different values of the remaining hyperparameter ($\beta$ in the leftmost). In general, high $\gamma$ and low $\epsilon$ gave the highest correlation with the gold standard LOO experiment, and $\beta$ was less significant in the toy model.

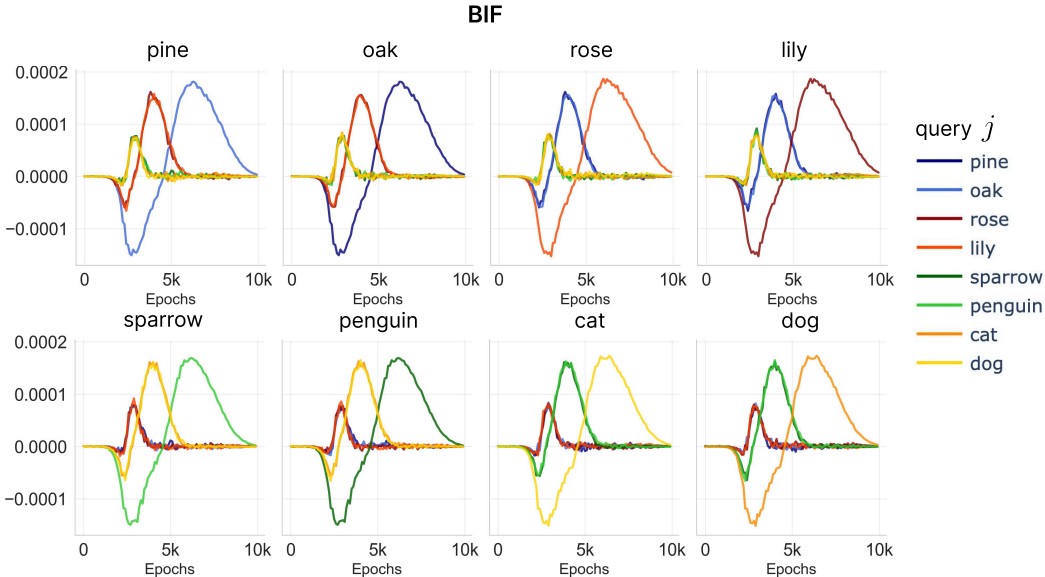

Figure 7: **BIF for perturbing each data point.** The subtitle indicates the perturbed data point.

at time $t$. Note that we do not use the global minimum $\boldsymbol{w}^*$ but the minimum at that specific measurement time $t$. Since the Hessian $\boldsymbol{H}$ of the overparameterized neural network is non-invertible due to multiple zero / near-zero eigenvalues, we take the approximation methods as in Koh & Liang (2020); Grosse et al. (2023). First is to add a constant dampening term $\gamma$ to the Hessian,

$$\tilde{\boldsymbol{H}} = \boldsymbol{H} + \gamma^*\mathbf{I}. \tag{17}$$

Suitable $\gamma$ enforces $\tilde{\boldsymbol{H}}$ to have positive eigenvalues. Grosse et al. (2023); Martens & Grosse (2015); Bae et al. (2022) used damped Gauss-Newton-Hessian (GNH), an approximation to $\boldsymbol{H}$ which linearizes the network's parameter-output mapping around the current parameters,

$$\boldsymbol{G} = \mathbb{E}[\mathbf{J}^\top \boldsymbol{H}\mathbf{J}] \tag{18}$$

$$\bar{\boldsymbol{H}} = \boldsymbol{G} + \lambda^*\mathbf{I}. \tag{19}$$

Using approximated damped-Hessian ($\tilde{\boldsymbol{H}}$) and damped GNH ($\bar{\boldsymbol{H}}$), we measure classical influence function over time (Eq. 16) with varying dampening constant $\gamma$.

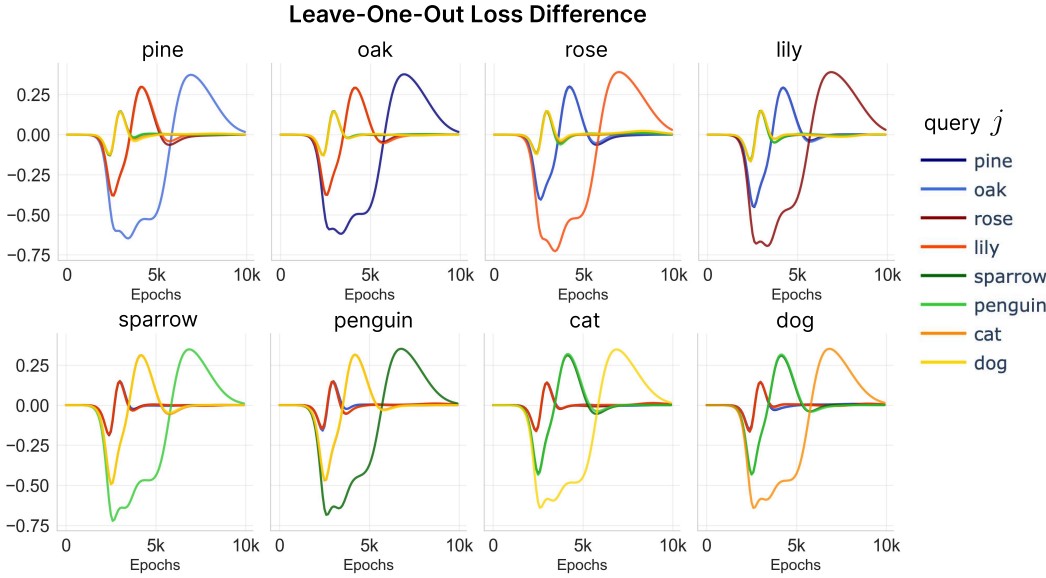

Figure 8: **Leave-One-Out (LOO) loss difference for perturbing each data point.** The title indicates the ablated data point.

The dampening constant $\gamma^*$ is scaled with $\alpha$, the maximum absolute value of all eigenvalues of the $\boldsymbol{H}$

$$\gamma^* = \gamma\alpha, \tag{20}$$

and we vary $\gamma$ in range of $[1e\text{-}3, 10]$ in logarithmic scale.

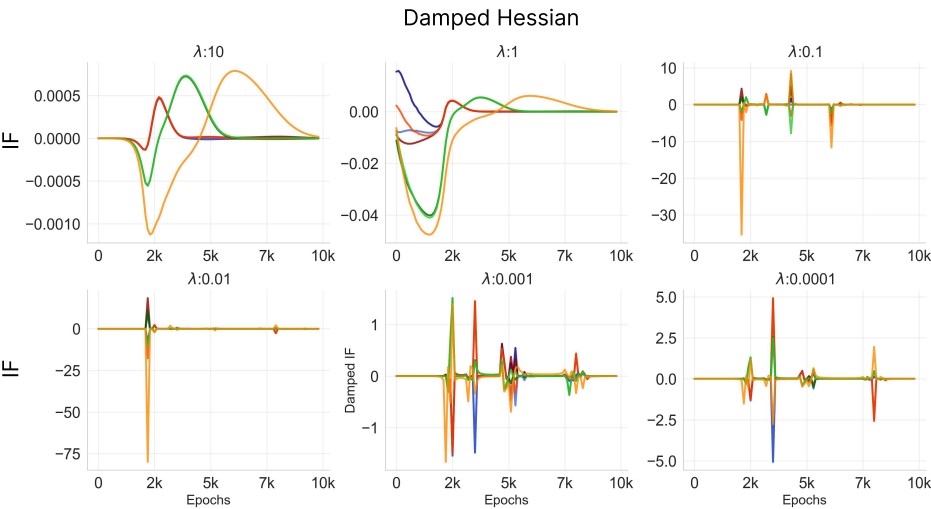

Figure 9: **Influence with damped Hessian approximation.** We measure the classical influence of the `dog` sample with damped Hessian approximation $\tilde{\boldsymbol{H}}$ with varying dampening constant $\gamma$.

## C.5   TIME-SPECIFIC ABLATION EXPERIMENT

Our observation that influence is a time-dependent function proposes another perspective – perturbing the same data point but at different training times would lead to non-identical influence. That is, the exact moment the network interacts with each data point matters. Here we probe this hypothesis with a simple retraining experiment in the toy model. We hypothesize that the measured influence

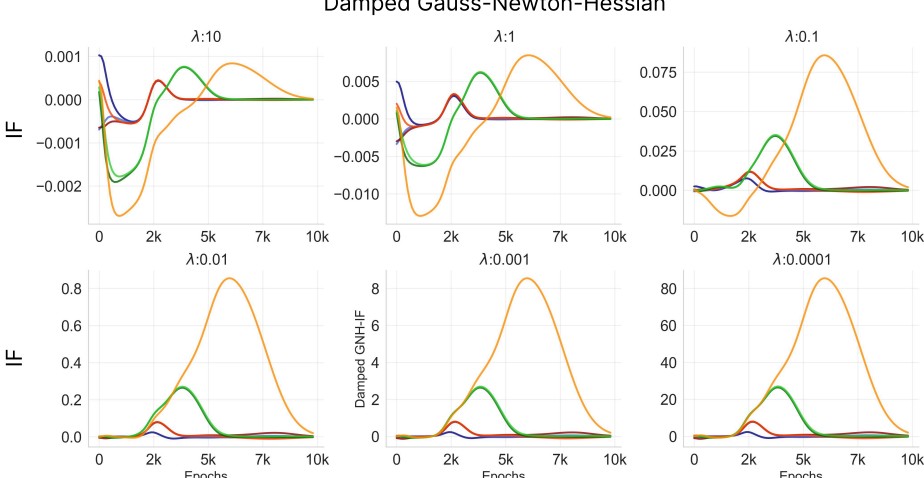

Figure 10: **Influence with damped Gauss-Newton-Hessian approximation.** We measure the classical influence of the `dog` sample with damped Gauss-Newton-Hessian approximation $\tilde{H}$ with varying dampening constant $\gamma$.

trajectory tells us which time point would be most critical for a model to learning each sample. For example, the influence of `dog` on `cat` makes a peak around $t = 6000$, which implies ablating `dog` around that time would be most helpful in learning `cat`. We probe this hypothesis with a retraining experiment with a brief ablation at different stages of learning.

In Figure 11, we ablate a data point `dog` during the duration of $D$ epochs from $t$ and show the integrated loss difference compared to the baseline (no ablation) during $D$. As we expected, the loss difference is the biggest around $t = 6000$, when the highest influence occurs, indicating ablating `dog` was helpful the most at this stage. Similarly, we observe the negative peak of integrated loss difference around $t = 3000$, which also matches the negative minimum in the influence measurement, implying that learning of the cat was the most detrimental at this time point when `dog` is ablated. We show that the most important stage changes with the query data that belongs to a different hierarchy, e.g., `sparrow`. Collectively, we strengthen our claim that influence over time shows us how a sample interacts with "what" data "when", and that it is often correlated with the structure of the data, such as hierarchy.

## C.6 ANALYTICAL INVESTIGATION OF TOY MODEL

In this section, we analytically derive the dynamics of the influence function in a deep linear neural network. First, we introduce deep linear neural network learning dynamics studied in Saxe et al. (2013; 2019b) and then we show the time dependency of the influence function and its relation to progressive learning of the hierarchical structure.

### DEEP LINEAR NEURAL NETWORK LEARNING DYNAMICS

Following the singular modes learning formulation in Saxe et al. (2019a; 2013), we can describe each data point $i$'s loss into the singular modes basis. We assume whitened input $X \in \mathbb{R}^{N \times D}$, $X^T X = I$ and output $Y \in \mathbb{R}^{N \times O}$. The input-output correlation and its singular value decomposition (SVD) become

$$\mathbf{C} := \Sigma_{yx} = Y^T X \tag{21}$$

$$SVD(\Sigma_{yx}) = \mathbf{U}\mathbf{S}\mathbf{V}^\top. \tag{22}$$

With squared error loss, individual data point at data $i$ is $\ell_i = \frac{1}{2}\|\mathbf{y}_i - \hat{\mathbf{y}}_i\|^2$. We project data $\mathbf{x}_i$ to $V$ space (object analyzer) and its component becomes $\varepsilon_{ik} = v_k^\top \mathbf{x}_i$, where $v_k$ is $k$th column of $V$. Similarly, we project $\eta_{ik} = u_k^\top \mathbf{y}_i$ and with rank $r = \text{rank}(\mathbf{C})$,

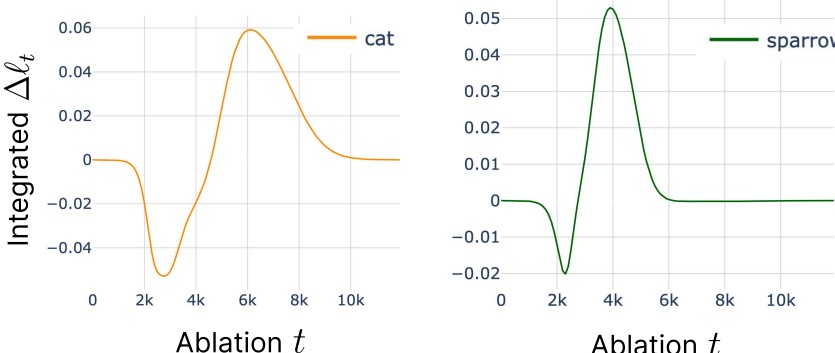

Figure 11: **Brief ablation of dog.** For each experiment, we ablate dog for duration $D = 100$ epochs starting from timepoint $t$. Starting time $t$ is sampled uniformly with an interval of 200 epochs. For each experiment, we integrate the loss difference to the baseline during $D$. We show the integrated loss of query (a) cat and (b) sparrow over different ablation windows. The most critical period of influence from the dog on the two samples is different. The higher level hierarchy (animals) sharing sample sparrow has an earlier critical point than the lower level hierarchy (mammals) sharing sample cat.

$$\ell_i = \frac{1}{2} \sum_{k=1}^{r} (\eta_{ik} - g_k \varepsilon_{ik})^2, \tag{23}$$

$g_k$ is a singular mode strength of mode $k$ of the network obtained from

$$\boldsymbol{W}(t) = \boldsymbol{W}_2(t)\boldsymbol{W}_1(t) = \boldsymbol{U}\boldsymbol{G}(t)\boldsymbol{V}^\top, \tag{24}$$

assuming the network and the data share the aligned right and left singular vectors $\boldsymbol{U}, \boldsymbol{V}$. This assumption holds with a small initialization and a small learning rate.

Now, we can compute the loss gradient with respect to the parameter space and decoupled mode space. In parameter space,

$$\nabla L_{\boldsymbol{W}_1} = \boldsymbol{W}_2^\top (\Sigma_{yx} - \boldsymbol{W}_2 \boldsymbol{W}_1) \tag{25}$$

$$\nabla L_{\boldsymbol{W}_2} = (\Sigma_{yx} - \boldsymbol{W}_2 \boldsymbol{W}_1)\boldsymbol{W}_1^\top. \tag{26}$$

We can substitute the terms in singular decomposed form,

$$\nabla L_{\boldsymbol{W}_1} = \boldsymbol{U}(\boldsymbol{S} - \boldsymbol{G})\boldsymbol{A}_1 \tag{27}$$

$$\nabla L_{\boldsymbol{W}_2} = \boldsymbol{A}_2(\boldsymbol{S} - \boldsymbol{G})\boldsymbol{V}^\top \tag{28}$$

$$\boldsymbol{W}_1 = \boldsymbol{R}\boldsymbol{A}_1\boldsymbol{V}^T, \boldsymbol{W}_2 = \boldsymbol{U}\boldsymbol{A}_2\boldsymbol{R}^T, \boldsymbol{A}_2\boldsymbol{A}_1 = \boldsymbol{G}. \tag{29}$$

The effective singular values from each layer $\boldsymbol{W}_{1,2}$ are $\boldsymbol{A}_{1,2}$ and $\boldsymbol{R}^T \boldsymbol{R} = I$, which eventually aligns the effective weights' singular vectors to that of the data correlation $\Sigma_{yx}$. In general, we cannot assume $a_{1k} = a_{2k}$ throughout learning, but the equality holds with the strictly balanced initialization. With the small initialization assumption, the first-order approximation of equality holds, which eventually allows us to write down the effective singular value trajectory throughout learning as in Saxe et al. (2019a).

Importantly, $\boldsymbol{G}(t)$ is a time-varying variable, evolving throughout the training that could be written in closed form with $s_k$ being the corresponding diagonal singular value of $\boldsymbol{S}$ from data covariance,

$$\boldsymbol{G}_{kk}(t) = \frac{s_k e^{2s_k t/\tau}}{e^{2s_k t/\tau} - 1 + s_k/\boldsymbol{G}_{kk}^0}, \quad \boldsymbol{G}_{kk}^0 = \boldsymbol{G}_{kk}(t=0), \tag{30}$$

and the model that learned the training data recovers

$$\boldsymbol{W}^* = \mathbf{UGV} = \mathbf{C}. \tag{31}$$

For convenience, we will drop the notation of time dependence in the rest of this section. For an end-to-end linear map $\boldsymbol{W} = \boldsymbol{W}_2\boldsymbol{W}_1 \in \mathbb{R}^{O \times D}$ define its representation in the data singular basis and the singular mode strength $g_k$,

$$\boldsymbol{G} := \boldsymbol{U}^\top \boldsymbol{W}\boldsymbol{V}, \qquad g_k := \boldsymbol{G}_{kk} = u_k^\top \boldsymbol{W} v_k. \tag{32}$$

For a data point $p = (x_p, y_p)$, we can write the coordinates in the *singular* basis

$$\gamma_p := \boldsymbol{V}^\top x_p \in \mathbb{R}^r, \qquad \eta_p := \boldsymbol{U}^\top y_p \in \mathbb{R}^r. \tag{33}$$

INFLUENCE FUNCTION IN TOY MODEL USING SINGLE DATA POINT PERTURBATION

Following Cook & Weisberg (1982); Koh & Liang (2020), we will derive *influence function*, that is, the change in model prediction with a targeted perturbation of training data. In the following, we will derive the response of original $\boldsymbol{U}, \boldsymbol{V}$ and $\boldsymbol{S}$ as a function of a single data point perturbation to derive the influence function in singular basis based on the derivation and results from Stewart & Sun (1990) on the first-order SVD perturbation. With those in hand, we can use the closed-form dynamics on Eq. 30 to get time-dependent dynamics.

First, we consider a perturbation by upweighting a single data point $p$ with factor $\varepsilon$. The perturbed cross-covariance becomes

$$\mathbf{C}_p(\varepsilon) = \boldsymbol{U}_p(\varepsilon)\boldsymbol{S}_p(\varepsilon)\boldsymbol{V}_p(\varepsilon)^\top = \sum_{p \neq j} \mathbf{y}_j\mathbf{x}_j^\top + (1+\varepsilon)\mathbf{y}_p\mathbf{x}_p^\top = \mathbf{C} + \varepsilon\mathbf{y}_p\mathbf{x}_p^\top. \tag{34}$$

Let $\mathbf{C}'$ be the first-order derivative of the data covariance with $\varepsilon$-upweighting of a single data point $p$,

$$\mathbf{C}_p' := \left.\frac{\partial \mathbf{C}_p(\varepsilon)}{\partial \varepsilon}\right|_{\varepsilon=0}. \tag{35}$$

With the perturbed dataset, the learned 2-layer linear network mapping at time $t$ becomes

$$\boldsymbol{W}_p(\varepsilon, t) = \boldsymbol{U}_p(\varepsilon)\boldsymbol{G}_p(\boldsymbol{S}_p(\varepsilon), t)\boldsymbol{V}_p^\top(\varepsilon), \tag{36}$$

$$\boldsymbol{G}_t(\boldsymbol{S}, t) = \mathrm{diag}(g_t(s_1), ...g_t(s_r)) \tag{37}$$

where $s$ is a singular value from diagonals of $\boldsymbol{S}$ and $g_t(\cdot)$ follows a closed form dynamics as given in Eq. 30. We assume that the network will have aligned singular basis with the perturbed data covariance, given in $\boldsymbol{U}(\varepsilon)$ and $\boldsymbol{V}(\varepsilon)$. For convenience, we will move the parameter $\varepsilon$ to a subscript and drop the perturbed data index $p$.

We apply product rule to $\mathbf{C}_\varepsilon'$,

$$\mathbf{C}_\varepsilon' = \boldsymbol{U}_\varepsilon'\boldsymbol{S}\boldsymbol{V}^\top + \boldsymbol{U}\boldsymbol{S}_\varepsilon'\boldsymbol{V}^\top + \boldsymbol{U}\boldsymbol{S}\boldsymbol{V}_\varepsilon'^\top. \tag{38}$$

We project the above perturbation to the singular mode coordinates,

$$\boldsymbol{Q}_\varepsilon := \boldsymbol{U}^\top \mathbf{C}_\varepsilon'\boldsymbol{V} = \boldsymbol{U}^\top\left(\boldsymbol{U}_\varepsilon'\boldsymbol{S}\boldsymbol{V}^\top + \boldsymbol{U}\boldsymbol{S}_\varepsilon'\boldsymbol{V}^\top + \boldsymbol{U}\boldsymbol{S}\boldsymbol{V}_\varepsilon'^\top\right)\boldsymbol{V}. \tag{39}$$

Since $\boldsymbol{U}, \boldsymbol{V}$ are orthonormal matrices,

$$\boldsymbol{U}^\top\boldsymbol{U} = \mathbf{I}, \; \boldsymbol{V}^\top\boldsymbol{V} = \mathbf{I} \tag{40}$$

and differentiating these orthonormality constraints with respect to the perturbation $\varepsilon$

$$\left.\frac{\partial}{\partial \varepsilon}\left(\boldsymbol{U}^\top\boldsymbol{U}\right)\right|_{\varepsilon=0} = \boldsymbol{U}_\varepsilon'^\top\boldsymbol{U} + \boldsymbol{U}^\top\boldsymbol{U}_\varepsilon' = 0. \tag{41}$$

$$\left.\frac{\partial}{\partial \varepsilon}\left(\boldsymbol{V}^\top\boldsymbol{V}\right)\right|_{\varepsilon=0} = \boldsymbol{V}_\varepsilon'^\top\boldsymbol{V} + \boldsymbol{V}^\top\boldsymbol{V}_\varepsilon' = 0. \tag{42}$$

Due to orthonormality, $\boldsymbol{U}_\varepsilon', \boldsymbol{V}_\varepsilon'^\top$ lie in the tangent space of $\boldsymbol{U}, \boldsymbol{V}^\top$. We can define

$$\boldsymbol{A} := \boldsymbol{U}^\top\mathbf{U}_\varepsilon', \; \boldsymbol{B} := \boldsymbol{V}_\varepsilon'^\top\boldsymbol{V}, \tag{43}$$

where the orthonormal constraints give

$$\mathbf{A}^\top + \mathbf{A} = 0, \quad \mathbf{B} + \mathbf{B}^\top = 0, \tag{44}$$

and $\mathbf{A}, \mathbf{B}$ become skew-symmetric, $\mathbf{A} = -\mathbf{A}^\top$, $\mathbf{B} = -\mathbf{B}^\top$. Intuitively, $\mathbf{A}, \mathbf{B}$ is equivalent to the in-subspace angular velocity (rotation rate) of the columns of $\mathbf{U}$ and $\mathbf{V}^\top$ due to the perturbation, respectively.

With this, we can rewrite Eq. 39

$$\mathbf{Q} = \mathbf{AS} + \mathbf{S}'_\varepsilon + \mathbf{SB}, \tag{45}$$

where $\mathrm{diag}(\mathbf{A}) = \mathrm{diag}(\mathbf{B}) = 0$ since $\mathbf{A}, \mathbf{B}$ are both skew symmetric.

Now, we solve Eq. 45 for $\mathbf{A}, \mathbf{B}$ and $\mathbf{S}'_\varepsilon$. First, we consider a case where the singular values $\mathrm{diag}(\mathbf{S})$ are non-degenerate. Due to the diagonal constraint of singular values, the response from the perturbation should also be diagonal for the singular values

$$\mathbf{S}'_{\varepsilon,ii} = \mathbf{Q}_{\varepsilon,ii} = \mathbf{U}_i^\top \mathbf{C}'_\varepsilon \mathbf{V}_i. \tag{46}$$

The response of the singular basis is obtained from Eq. 43,

$$\mathbf{U}'_\varepsilon = \mathbf{UA}, \quad \mathbf{V}'^\top_\varepsilon = \mathbf{BV}^\top, \tag{47}$$

where

$$\mathbf{A}_{ij} = \begin{cases} \dfrac{s_j \mathbf{Q}_{ij} + s_i \mathbf{Q}_{ji}}{s_j^2 - s_i^2}, & i \neq j, \\ 0, & i = j, \end{cases} \tag{48}$$

$$\mathbf{B}_{ij} = \begin{cases} \dfrac{s_i \mathbf{Q}_{ij} + s_j \mathbf{Q}_{ji}}{s_j^2 - s_i^2}, & i \neq j, \\ 0, & i = j. \end{cases} \tag{49}$$

When $\mathbf{S}$ has degenerate singular values or a near-zero gap, above is ill-defined as the denominator $s_j^2 - s_i^2 = 0$. In this case, we apply block perturbation, treating the whole invariant subspace (sharing the degenerate singular values) all at once.

We take the sets of the degenerate singular values $\{s_b : s_j = s_b, \forall s_j \in \mathrm{diag}(\mathbf{S})\}$, define the block coupling by projecting the perturbation $\mathbf{C}'$ on this block,

$$\bar{\mathbf{Q}}_\varepsilon := \bar{\mathbf{U}}^\top \mathbf{C}'_\varepsilon \bar{\mathbf{V}}, \tag{50}$$

where $\bar{\mathbf{U}}$ and $\bar{\mathbf{V}}^\top$ are sets of the degenerate singular values corresponding to sets of columns of $\mathbf{U}$ and $\mathbf{V}$. It can be further decomposed into a symmetric and a symmetric-skewed part,

$$\bar{\mathbf{M}}_\varepsilon = \frac{1}{2}(\bar{\mathbf{Q}}_\varepsilon + \bar{\mathbf{Q}}_\varepsilon^\top), \quad \bar{\mathbf{K}}_\varepsilon = \frac{1}{2}(\bar{\mathbf{Q}}_\varepsilon - \bar{\mathbf{Q}}_\varepsilon^\top), \quad \bar{\mathbf{M}}_\varepsilon + \bar{\mathbf{K}}_\varepsilon = \bar{\mathbf{Q}}_\varepsilon. \tag{51}$$

Due to diagonality constraints of singular values and zero-diagonals in the skew-symmetric part $\bar{\mathbf{K}}_\varepsilon$, the first-order singular value consistency reduces to $\bar{\mathbf{M}}_\varepsilon = \bar{\mathbf{S}}'_\varepsilon$. To have diagonal singular value matrix $\bar{\mathbf{S}}'_\varepsilon$, we choose $\mathbf{R}$ such that

$$\bar{\mathbf{M}}_\varepsilon = \bar{\mathbf{S}}'_\varepsilon = \mathbf{R}^\top \Lambda \mathbf{R}, \tag{52}$$

where $\mathrm{diag}(\Lambda)$ becomes the first-order splits within the degenerate block with preferred direction defined by $\mathbf{R}$. Skewed part $\bar{\mathbf{K}}_\varepsilon$ sets the in-block angular velocity,

$$\bar{\mathbf{A}} = \bar{\mathbf{B}} = \frac{1}{2s_b}\bar{\mathbf{K}}_\varepsilon. \tag{53}$$

Then,

$$\bar{\mathbf{S}}'_{\varepsilon,ii} = \Lambda_{ii}, \quad \bar{\mathbf{U}}'_\varepsilon = \bar{\mathbf{U}}\bar{\mathbf{A}}, \quad \bar{\mathbf{V}}'_\varepsilon = \bar{\mathbf{V}}\bar{\mathbf{B}}. \tag{54}$$

Now we are equipped with $\boldsymbol{U}_\varepsilon', \boldsymbol{V}_\varepsilon', \boldsymbol{S}_\varepsilon'$ described by original we define influence on $\boldsymbol{W}(t, \varepsilon)$ from the perturbation on a data point $p$ at fixed time $t$ by applying derivative on Eq. 36,

$$\mathcal{I}_p^W := \left. \frac{\partial \boldsymbol{W}}{\partial \varepsilon} \right|_{\varepsilon=0} = \boldsymbol{U}_\varepsilon' \boldsymbol{G}(\boldsymbol{S}, t) \boldsymbol{V}^\top + \boldsymbol{U} \boldsymbol{G}(\boldsymbol{S}_\varepsilon', t) \boldsymbol{V}^\top + \boldsymbol{U} \boldsymbol{G}(\boldsymbol{S}, t) \boldsymbol{V}_\varepsilon'^\top \tag{55}$$

$$= \mathbf{U}\mathbf{A}\boldsymbol{G}(\boldsymbol{S}, t)\boldsymbol{V}^\top + \boldsymbol{U}(\mathrm{diag}(g_t'(s)) \odot \mathrm{diag}(\boldsymbol{S}_\varepsilon')))\boldsymbol{V}^\top + \boldsymbol{U}\boldsymbol{G}(\boldsymbol{S}, t)\mathbf{B}\boldsymbol{V}^\top. \tag{56}$$

We can further derive influence on measurable $\Phi_i$, squared loss $\frac{1}{2}\|\phi_i\|^2$ with residual of measured data point $i$ $\phi_i$ using the chain rule,

$$\mathcal{I}(z_p, \Phi_i) = \left\langle \phi_i x_i^\top, \frac{\partial \boldsymbol{W}}{\partial \varepsilon} \right\rangle = \phi_i^\top \left( \frac{\partial \boldsymbol{W}}{\partial \varepsilon} x_i \right) = \phi_i^\top \boldsymbol{U}(\mathbf{A}\mathbf{G}_t + \boldsymbol{G}_{\varepsilon,t}' + \boldsymbol{G}_t \mathbf{B})\boldsymbol{V}^\top x_i, \tag{57}$$

where $z_p$ is the perturbed data point. This perturbation is expected to hold with a small perturbation $\varepsilon \approx 0$. We use down-weighting $\varepsilon = -0.1$ to approximate the ablation effect in Figure 3.

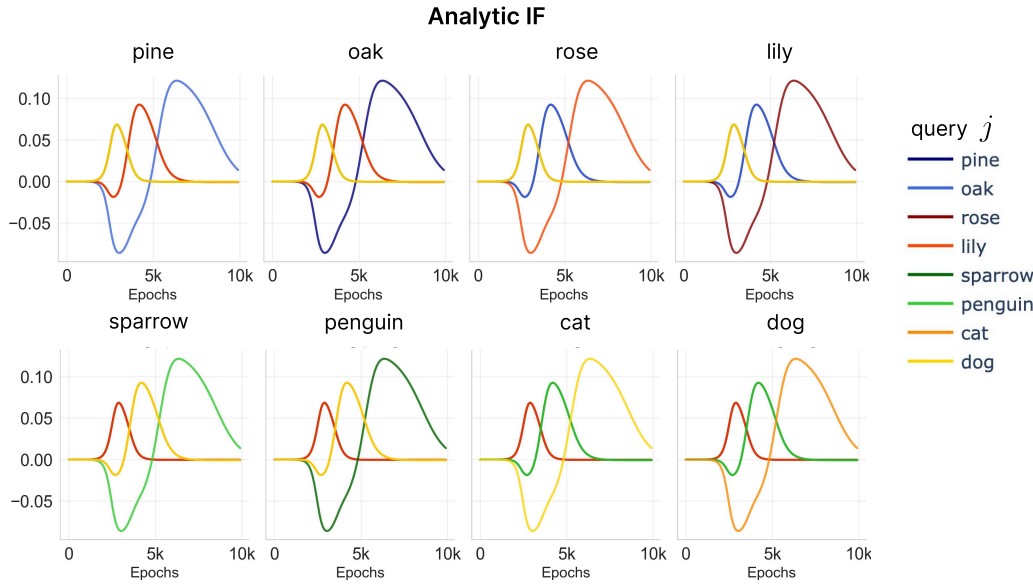

Figure 12: **Analytical loss difference from perturbing each data point.** The title indicates the perturbed data point ($\varepsilon = -0.1$).

# D LANGUAGE MODELS

This section discusses additional experimental details for the language model experiments.

## D.1 STRUCTURAL TOKEN CLASSIFICATION

Here, we present additional details and further experiments conducted to investigate the development of influence with respect to how tokens structure text. These experiments were conducted using the 14 million parameter Pythia model. Tokens are generated using the same tokenizer as the Pythia models use in order to be able to use these tokens with the model suite.

### D.1.1 TOKEN CLASSES

Tokens are classified based on their role in structuring text. The classes we used are based on the classes used by Baker et al. (2025). Figure 13 demonstrates these classes graphically, with tokens outlined in bold indicating class membership, and distinct colors representing distinct classes.

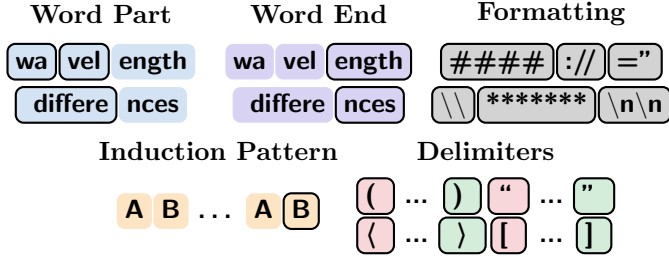

Figure 13: **Examples of structural classification of tokens.** Bold outline indicates class membership.

The classes are as follows:

**Word Part**  Tokens that constitute a part of a word that is not the end of the word. This class excludes tokens that represent entire words or entire words with the addition of additional characters (e.g. a space and an entire word).

**Word End**  Tokens that constitute the ending of a multi-token word. This class also excludes tokens that represent entire words and entire words with the addition of additional characters.

**Formatting**  Tokens that are generally used to format text including newlines and repeated characters.

**Induction Pattern**  Tokens that appear in a context in which the token is preceded by another particular token, and the same token has been preceded by the same other token earlier in the context.

**Left Delimiters**  Tokens that are generally followed by an equivalent right delimiter with interceding text are considered to occupy a distinct scope.

**Right Delimiters**  Tokens that close the scope opened by a paired left delimiter.

### D.1.2  RMSPROPSGLD HYPERPARAMETERS

Table 2: Summary of hyperparameter grid search for BIF experiments on Pythia 14M.

| Hyperparameter | Range |
|---|---|
| $n$ (number of sequences) | 600 |
| $J$ (sequence length) | 55 |
| $\epsilon$ (step size) | $1e-6$ |
| $n\beta$ (inverse temperature) | $\{256, 1024\}$ |
| $\gamma$ (localization strength) | $\{500, 1000\}$ |
| $m$ (batch size) | 64 |
| $C$ (number of chains) | 4 |
| $T$ (chain length) | 200 |
| burn in steps | 0 |
| $k$ (nearest neighbors) | 30 |

We found that the overall dynamics expressed by the BIF did not vary substantially among hyperparameters chosen in this setting. Plots in the main paper were selected based on the parameters that optimized the average class recall on the KNN experiment which is discussed in section Appendix D.1.3

### D.1.3  SAME-CLASS PREDICTION WITH BIF K-NEAREST NEIGHBORS

One way to investigate the ability of the BIF to capture what sorts of text structure a model has learned is to use it to predict which structural class a token belongs to based on the classes that token shares high influence with.

We query the predictive capacity of token-level influences using a simple nearest-neighbors approach. The predictive model is as follows:

$$y_i(x_i) = \text{argmax}_a \frac{1}{r_a(x_i)} \sum_{x_j \in m_i} \mathbf{1}_a(x_j),$$

where $\mathbf{1}_a$ is the indicator function for elements of class $a$, and $r_a$ is the token-dependent class frequency in the dataset:

$$r_a(x) = \frac{1}{\sum_{x'} \mathbf{1}_{\{x\}^c}(x')} \sum_{x'} \mathbf{1}_{\{x\}^c}(x') \mathbf{1}_a(x'),$$

with $\{x\}^c$ denoting the set of all tokens that are not $x$. Tokens are thus predicted to have the majority label of their corresponding top-influence set $m_i$, adjusted for class rates overall.

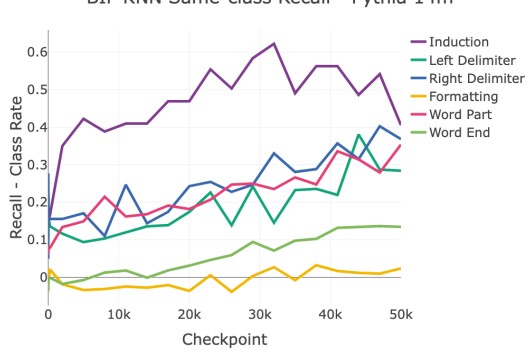

Figure 14: **KNN BIF predictions change over time.** Using the BIF in order to select the top 30 highest influence tokens and taking the majority class from among them (adjusted for class rates), we plot the recall minus the class rate for each class. This means that 0 represents the random baseline. We see that the BIF selects for in-class tokens above random for all categories by the end of training, indicating that it is sensitive to the class structure learned by the model.

Figure 14 shows the results of this experiment. We observe that the classes of high-influence other tokens act as a better than random predictor for all classes by the end of training, with recall generally improving as the model progresses through training.

### D.2 PER-TOKEN INFLUENCE DYNAMICS

We presented a coarse-grained analysis of aggregated influence between different classes of syntactic tokens in Figure 5. Here, we additionally measured intra-class BIFs over training phases to capture more fine-grained influences between token classes. We focus on right and left delimiters classes, where we can make finer categorization of tokens into parentheses, curly brackets, square brackets, and angle brackets. In other words, we make finer categories of delimiters, including types, not only left and right directionality. In total, there are 8 classes (2 directions times 4 types). We exclude tokens that have more than one of different token types (e.g., `) ]` ). We measure influence dynamics on Pythia-14m model throughout the training for each reference token type. The hyperparameters for BIF measurement are kept the same as in Figure 5.

In Figure 15, we observe that different subgroups of tokens impose different magnitudes and signs of influence, which again can change over time. We see a few interpretable features. For example, we see that the same subtypes of brackets can assist learning as training goes by, even if the directionality is different (observed for curly brackets, square brackets, and angle brackets but not for parentheses), which might be due to the paired nature of the bracket signs.

### D.3 INFLUENCE DYNAMICS OF INDUCTION HEAD FORMATION

Our initial experiments with the public Pythia checkpoints (Biderman et al., 2023) suggest a signal for the learning of induction patterns. However, the checkpoint frequency was too coarse in the vicinity

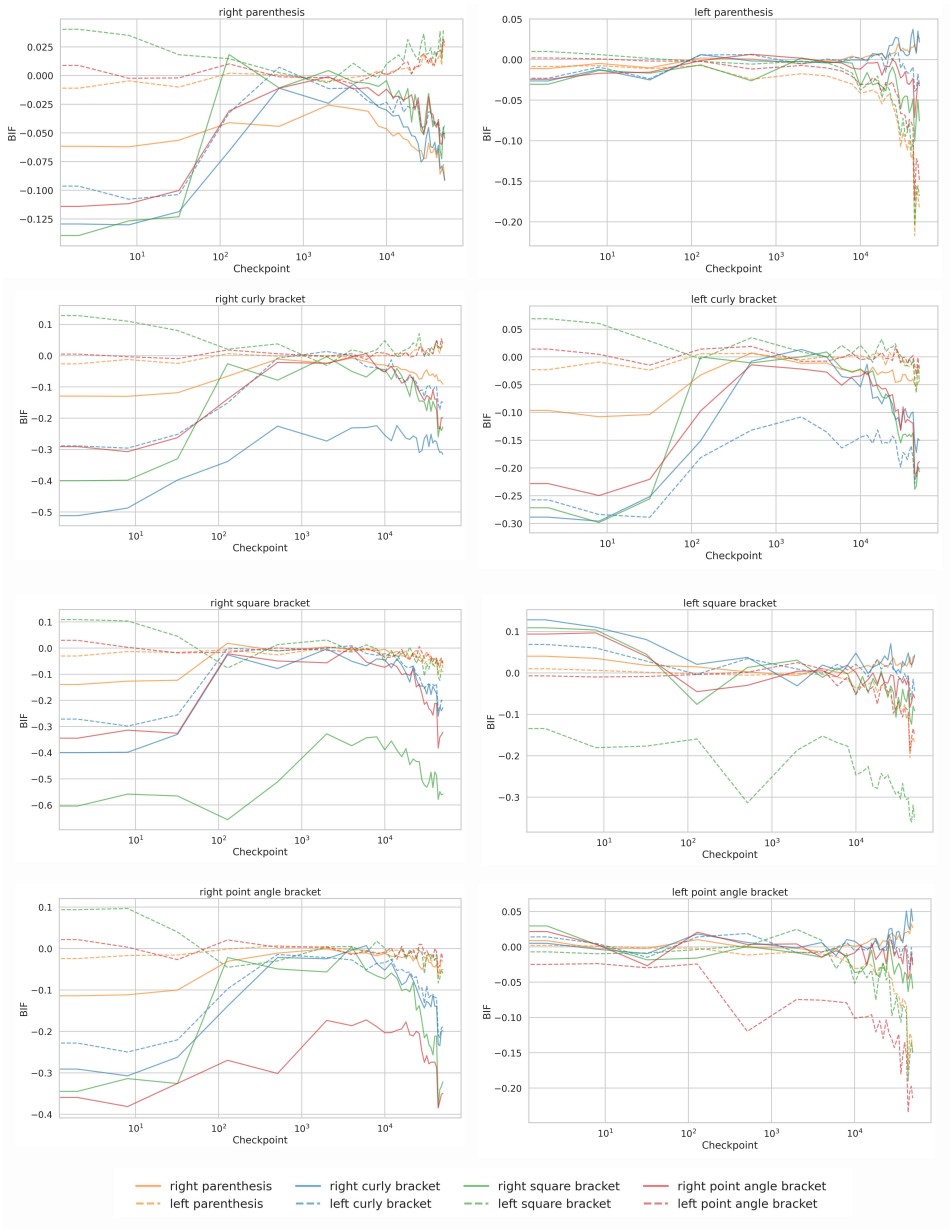

Figure 15: **Intra-classes (delimiters subtype) BIFs in Pythia-14M.** We plot BIFs between more fine-grained syntactic token classes of delimiters.

of checkpoints 1k to 10k to observe the fine-grained dynamics of this developmental transition. To investigate this critical phase in more detail, we trained a Pythia-14M model from scratch, saving checkpoints every 100 steps for the first 20,000 training steps. Though we train on the same Pile data (Gao et al., 2021), it is in a different order and on different hardware, which means this training run is not the same as the original Pythia-14M training run.

We constructed a synthetic dataset with sequences where the second half repeats the first, e.g., *Sequence 1*: `A B ... A B` and *Sequence 2*: `C D ... C D`. Sequences were constructed as to not share tokens. The first half of each sequence is made up of random tokens, and so the only structure that can be used for prediction is the repetition of the second half of the context.

During SGLD we *collect* losses on this dataset, but *sample* using the Pile, meaning that loss on these synthetic samples does not affect our sampling trajectory. We compute the BIF matrices at each checkpoint of our homemade Pythia-14M and then look at the mean correlation between the different parts of our sequence (Namely repeated tokens with repeated tokens from other sequences, non-repeated tokens with non-repeated tokens from other sequences, and non-repeated tokens with repeated tokens from other sequences).

The results are shown in Figure 16. The influence between tokens in the *repeated* segments of each sequence (top panel, blue line) undergoes a sudden, large increase, peaking and then stabilizing. Meanwhile, the BIF between non-repeated segments or across non-repeated and repeated segments shows no such change.

Simultaneously, we measure the "induction score" of the model's attention heads—a standard metric from mechanistic interpretability that quantifies how strongly a head implements the induction algorithm (Olsson et al., 2022). As shown in the bottom panel, the induction score for the heads that become the "induction heads" begins to rise right before the BIF increases between the repeated groups.

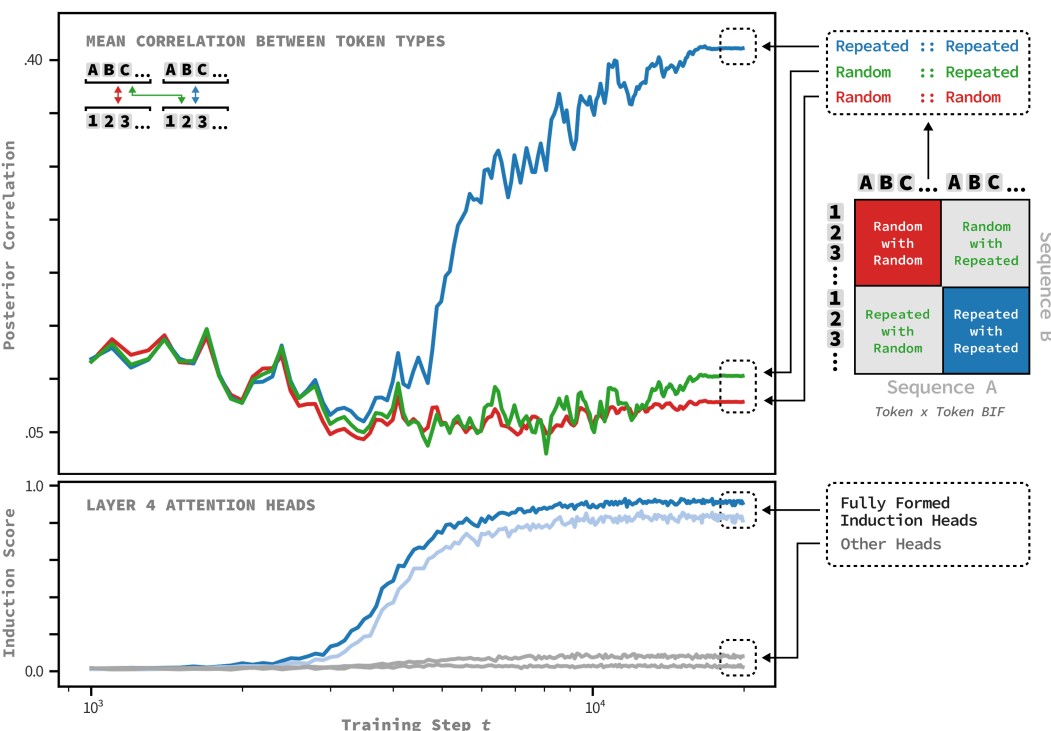

Figure 16: **Influence dynamics during induction head formation.** We trained a small transformer from scratch with high-frequency checkpointing to capture the formation of induction heads. (Top) The normalized BIF between corresponding tokens in repeated sequences (blue) shows a sharp increase and peak, while correlations between non-repeated samples (red) or between non-repeated and repeated segments (green) remain low. (Bottom) The induction score for the attention heads that become induction heads rises together with the posterior correlation, while other heads remain inactive.

## D.4 INTERVENING IN INDUCTION HEAD FORMATION

To validate the causal implications of stagewise influence, we conducted an intervention experiment targeting the acquisition of induction (Olsson et al., 2022). Following the results from the previous section that show that the influence of induction patterns can vary significantly over training, we predict that upweighting the same subset of data should have different effects on model development depending on *when* that upweighting occurs. In particular, we expect upweighting induction samples

will lead to faster or stronger induction head formation if these samples are upweighted during periods when the average influence between tokens representing an induction pattern is more strongly negative, indicating that losses across induction tokens are correlated and induction is being learned

**Setup.** We trained a 3-million parameter transformer on a dataset consisting of 13 million sequences from The Pile for a total of 15,000 steps. To isolate the effect of timing, we created five experimental runs where we upweighted the loss for tokens belonging to an induction pattern (as defined in Appendix D.1.1) by a factor of 4 during specific 500-step windows: 0–500, 1k–1.5k, 2k–2.5k, 3k–3.5k, 4k–4.5k, and 5k–5.5k. We compared these against a control model trained with no upweighted loss. All models shared identical initialization and data ordering; thus, deviations in dynamics are strictly attributable to the timing of the intervention.

We track the formation of the induction circuit using two methods: the induction score metric described in Appendix D.3, and the average BIF between induction tokens (excluding same-token pairs) on the unweighted baseline model for the first 6k training steps. We compare the induction scores for each model's highest-scoring head in Figure 17, and evaluate how induction learning dynamics are sensitive to the timing of BIF learning.

**Results.** We find that the BIF between induction tokens is highly informative of how the timing of the upweighting window alters the developmental trajectory of the induction circuit. This confirms that the influence function can be used to measure stagewise data sensitivity during model training. In the control setting, the induction circuit emerges gradually, with the induction score peaking relatively late in the training process at timestep 9k. Correspondingly, we observe that the BIF between induction tokens (on the synthetic dataset from the preceding section) is generally decreasing during this regime, indicating increasingly correlated learning on induction tokens. The BIF decreases relatively gradually for the first 2k tokens after which we observe a more rapid decline. Upweighting induction tokens during early training stages while the BIF is more positive (indicating interference) leads to delayed learning or completely disrupts learning altogether. There is an intermediate point where upweighting in the 2k–2.5k window leads to a similar learning dynamic to the baseline, though the induction score remains higher for longer after peaking. Intervening during stages beyond this point, when the BIF is decreasing rapidly, leads to accelerated learning, with induction scores rapidly spiking during the intervention window for the 4k–4.5k and 5k–5.5k interventions.

These results provide causal evidence for our stagewise framework. Upweighting a data pattern outside the phase where the model is receptive to it can prevent learning or yield developmental delays, whereas upweighting it inside the critical window can accelerate structure formation. We find that the BIF can be used to measure these dynamics, with a more positive BIF indicating that learning will be delayed or disrupted while a more negative BIF indicates faster and stronger effects of token weighting.

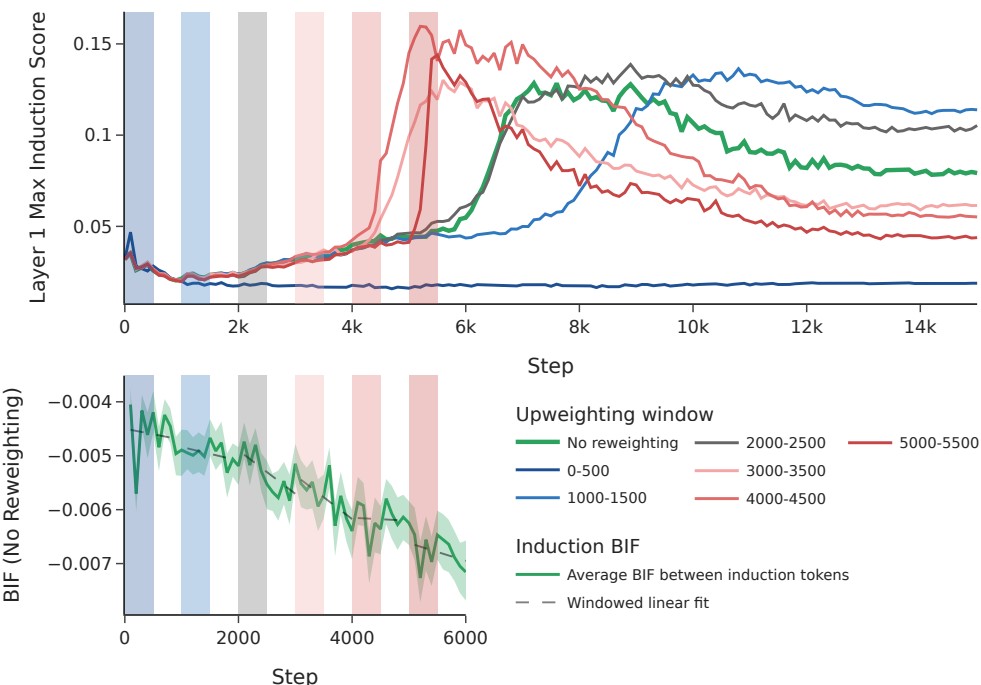

Figure 17: **Stagewise intervention accelerates induction head formation.** A comparison of the maximum induction score across Layer 1 heads for each intervention window (top). The baseline model (top, green) naturally forms induction heads at step ∼7k. Upweighting induction data during the critical developmental windows, indicated by red lines (top) and windows (top & bottom), significantly accelerates this capability, causing heads to form as early as step 5k. In contrast, interventions applied too early fail to produce stable induction heads or delay induction learning, see blue lines (top) and windows (top & bottom). There is an intermediate window where there is little effect of upweighting on learning dynamics (gray). Comparing the timing of these windows to the BIF of the baseline model (green, bottom), we observe that decreasing BIF indicates faster learning, as predicted. The shaded region is a 90% CI of the 100-sample block-bootstrap from the approximate posterior estimated via Algorithm 1.

