# OpenReview forum: "Influence Dynamics and Stagewise Data Attribution"
_ICLR.cc/2026/Conference — ICLR 2026 Poster_

### Official Review · Reviewer_CzFc · 2025-10-27

**Soundness:** 3
**Presentation:** 3
**Contribution:** 3
**Rating:** 6
**Confidence:** 3

**Summary:**

The paper introduces a stage-wise influence framework to track how data influence evolves throughout training. It predicts non-monotonic behavior and validates these in both a toy model and large language models (LLMs).

**Strengths:**

1. It introduces a novel framework of *stage-wise influence*, enriching the understanding of data attribution beyond static snapshots.
2. Makes clear predictions and uses Bayesian Influence Functions to measure influence over time, confirming them in a toy model and in LLMs.

**Weaknesses:**

1. Empirical verification depends on BIF and SGLD:
Validation relies on BIF, which requires sampling a local posterior via RMSProp-SGLD around each checkpoint, while most models are trained with Adam-family optimizers, raising questions about stage-wise behavior under Adam.

2. Computational overhead:
BIF uses Monte Carlo sampling, which is costly; complexity/scalability is not thoroughly analyzed, so practicality at scale remains uncertain.

3. Limited experimental scope:
Experiments emphasize inter-class patterns (to align with Sec. 2.3), but broader claims about *when* data are influential and *how* they shape internals would benefit from intra-class analyses and more mechanistic evidence

**Questions:**

1. Figure 3: BIF and analytical IF resemble LOO but peak at different times. Why don’t the maxima align?
2. Figure 7: The influence curve between “Dog” and all plant classes looks identical; the influence curve of “lily” and all animal classes mirrors that shape. Is this expected symmetry or some artifacts?
3. Pointer: The text says the RMSProp-preconditioned SGLD sampler is “introduced in Section 2.2,” but the *details* are in Appendix B.

---

> ### Author Response · Authors · 2025-11-22
>
> Thank you for your review. We are pleased that you recognized the novelty of our stagewise framework and the soundness of our theoretical predictions. We appreciate your constructive feedback on the experimental scope, which has motivated us to include additional analyses.
>
> **W1: Empirical Verification (BIF/SGLD vs. Adam)**
>
> *Critique: Validation relies on sampling via RMSProp-SGLD, but models are trained with Adam. This raises questions about validity.*
>
> We wish to clarify an important distinction: the choice of sampling method is independent of the training optimizer.
>
> The BIF measures a property of the local posterior around checkpoint $w\_t^\*$. This loss landscape is the same regardless of whether you used Adam, SGD, or any other optimizer to reach that checkpoint. It’s true that the training optimizer determines which checkpoint you reach at time $t$, but once you’re there, the posterior is an intrinsic property of that location.
>
> We use RMSProp-SGLD simply because it is an efficient sampler for this geometry. It does not need to match the training trajectory because we are not attempting to replicate training; we are capturing the local “shape” of the influence at a given moment in time.
>
> **W2: Computational Overhead and Scalability**
>
> *Critique: BIF uses Monte Carlo sampling; complexity/scalability is not thoroughly analyzed.*
>
> This is a valid point. We have added a new paragraph at the end of Appendix B to explicitly discuss scalability tradeoffs. While SGLD does incur a sampling cost, the BIF offers a specific advantage for our setting of trajectory analysis because it bypasses the expensive fitting stage of methods like EK-FAC.
>
> Scalability and complexity is a challenge common to all influence function methods, and we appreciate your suggestion to bring this explicitly into our manuscript. Nevertheless, we hope our contribution on framing influence dynamics and stage-wise TDA stands as it is.
>
> **W3: Limited Experimental Scope (Intra-class & Mechanism)**
>
> *Critique: Broader claims would benefit from intra-class analyses and more mechanistic evidence.*
>
> Thank you for your comment. We agree that additional experiments would strengthen our claims and have added additional traces of within-class per-token influence trajectories to Appendix D.2 and added an interventional experiment to study whether our framework is predictive in the setting of induction formation. Please see our top-level comment for more details.
>
> **Questions.** Thank you for the questions.
>
> * **Q1: Figure 3 (Maxima misalignment):** Thank you for spotting this. This was a labeling error due to a mismatch in the hyperparameters between the analytical plot and the simulation, which we have now corrected.
> * **Q2: Figure 7 (Symmetry):** You are correct; this is an expected symmetry. The toy dataset is constructed with a strict hierarchy that makes the influence patterns for symmetric branches (e.g., plants vs. animals) mirror each other.
> * **Pointer**: Thank you – we have fixed the reference to Appendix B.
>
> **Final comments.** Thank you again for your constructive feedback. We hope our additional experiments and clarification strengthen our paper and contribution.

---

> > ### Comment · Reviewer_CzFc · 2025-11-25
> >
> > Thank you for the additional explanation and experiments. I appreciate the additional paragraph in Appendix B. However, I remain concerned about the computational overhead. The added discussion is still quite vague, and no explicit asymptotic or time analysis is provided.
> >
> > In fact, I do not fully agree with the rebuttal’s claim that BIF “bypasses” the expensive fitting stage of methods like EK-FAC. From Algorithm 1, each checkpoint requires $C*T$ full data forward, e.g., 4×200 for Pythia. If these hundreds or thousands of forward passes are repeated at every checkpoint, the cumulative cost may even exceed the fitting stage of EK-FAC. This makes the practical scalability still uncertain.
> >
> > However, I understand the choice of BIF given its flexibility to be applied at any training stage, which is important for stage-wise influence analysis. I also acknowledge the paper’s contribution in framing influence dynamics and stage-wise TDA. Overall, I would like to maintain my score, which leans toward acceptance.

---

> > > ### Author Response · Authors · 2025-11-28
> > >
> > > Thank you for your reply. We would like to clarify your point about the BIF’s scalability.
> > >
> > > First, we acknowledge that studying influence over time is inherently more expensive than static, endpoint-only analysis. This is a fundamental cost of moving from static to dynamic interpretability, regardless of the specific method used.
> > >
> > > Second, regarding the comparison with EK-FAC: To faithfully track influence dynamics (i.e., how influence changes at different training stages), we have to compare *at each checkpoint* individually, which requires rerunning the fitting step at each checkpoint for EK-FAC. The reviewer mentions “full data forward.” It is important to clarify that while the SGLD updates utilize mini-batches from the full distribution, the covariance estimation (the “scoring” passes) only requires forward passes over the specific attribution subset ($n\_{attribution}$) and query set we wish to analyze, not necessarily the entire training corpus.
> > >
> > > Our current conclusion on the BIF’s scalability is based on [1], which performed a detailed scalability comparison of the BIF against EK-FAC. We hope our explicit analysis below of the per-checkpoint costs (drawing on [1]) further clarifies your concern. For convenience we consider $n=n\_{attribution}=q$:
> > >
> > > * **EK-FAC:**
> > >   * EK-FAC has an up-front computational cost for fitting the Kronecker factors that scales linearly with the size of a large representative set of samples (not the full training dataset, but typically much larger than the attribution set) and cubically in each layer's width: $O(d\_{l}^3)$ per block, given $d\_{l}$ as the width of the layer. This cost amortizes over all subsequent queries.
> > >   * After fitting, scoring $n$ queries against $n$ training points requires $n^2$ gradient computations and vector products.
> > > * **BIF Cost:**
> > >   * The BIF bypasses this cubic fitting phase entirely. There is no need to fit factors.
> > >   * The BIF requires drawing $CT$ SGLD draws, which invoke a time cost very similar to $CT$ equivalent training steps, with a time complexity $O(CTd_\\textrm{total})$. At each draw, we compute forward passes over $n$ samples in an attribution set, for a total cost of $O(CTnd\_{\\textrm{total}})$. Though the forward passes are individually less expensive than backwards passes, there are many more forward passes, so this dominates the total costs of the BIF.
> > >   * After computing these losses, there is an additional empirical covariance estimation step over the $CTn$ losses, but this calculation is a vanishing contribution.
> > >
> > > In our experiments (and likely in most practical interpretability settings), we analyze a targeted subset of training data (e.g., $n\_{attribution} \= 400$ samples in Section 4.3). In this regime, the tradeoffs favor the BIF. If we were to significantly increase $n\_{attribution}$ (e.g., to attempt to measure influence over the full dataset), then the tradeoff would indeed shift towards favoring EK-FAC.
> > >
> > > If you believe that this paper would benefit from a more explicit analysis along the lines of the above, beyond what has already been added in, we would be happy to take this feedback and extend the section.
> > >
> > > Thank you again for your time, and we appreciate your quick response.
> > >
> > > [1] Kreer et al. "Bayesian Influence Functions for Hessian-Free Data Attribution" (2025) arXiv:2509.26544

---

### Official Review · Reviewer_LhWa · 2025-10-30

**Soundness:** 2
**Presentation:** 3
**Contribution:** 3
**Rating:** 4
**Confidence:** 3

**Summary:**

This paper argues for a position that training data attribution should be done dynamically, rather than statically at the end of the training. Inspired by the singular learning theory, the paper predicts a non-monotonic influence trajectories with sign flips and sharp peaks. The paper then simulates several scenarios to demonstrate their prediction.

**Strengths:**

The stage-wise attribution problem raised by the authors is interesting and well-presented. I am convinced that the same data point might be influential to other data in different ways at different stages of the training.

**Weaknesses:**

While the stage-wise framing is compelling, simply listing the most influential samples is of limited use. To establish real value, attribution should enable actionable interventions that beat a naive one-shot final stage attribution baseline. I strongly suggest that the authors please dedicate more space to what stage-wise attribution concretely enables. As of the current draft, the support is largely heuristic; adding such experiments would substantially strengthen the practical case.

**Questions:**

1. As the authors also brought up in the discussion section, there is a class of unrolling-based attribution methods. Can the authors propose some further discussions about these methods? Does the sign flip somehow imply that these methods are inaccurate, since the middle stage influences are canceling each other?
2. The current motivation/theoretical analysis is conducted solely based on BIF, which is a relatively new paper on this field. I wonder can the authors provide motivation in a more multifaceted manner?

---

> ### Author Response · Authors · 2025-11-21
>
> We thank you for your review. We are very glad you found our framing of “stagewise” attribution interesting and convincing. We appreciate your constructive push for more concrete validation, which has led us to include additional experiments.
>
> **W1: Actionable Interventions vs. Naive Baseline**
>
> *Critique: Simply listing influential samples is of limited use. The paper needs actionable interventions that beat a naive baseline to establish real practical value.*
>
> We agree that demonstrating actionable interventions would strengthen the paper and appreciate this suggestion. We address this in two ways:
>
> 1. **Existing Interventions (Toy Model):** We wish to highlight that Appendix C.5 (Figure 11\) already demonstrates this specific utility. By ablating samples *only* during the high-influence window identified by the BIF, we showed that the model is significantly more sensitive to data removal during these critical periods than at other times. We have improved the forward reference to these experiments in Sec. 3 (second to last paragraph) to make it easier to find.
> 2. **New LLM Interventions:** To demonstrate this in a natural-data setting, we have added a new set of stagewise intervention experiments (described in our top-level comment and in the new Appendix D.4). We confirm that upweighting induction tokens during the critical window identified by the BIF accelerates induction head formation significantly more than upweighting them before the window. This effect cannot be predicted from a static analysis at the end of training.
>
> **Q1: Implications for Unrolling-based Methods**
>
> *Question: Does the sign flip imply that unrolling-based methods (which aggregate influence over time) are inaccurate due to cancellation?*
>
> This is an interesting question. We don't have a complete answer, but we can speculate.
>
> We believe influence dynamics can indeed render unrolling methods misleading (just like they can with classical influence function approximations). It is probably possible to design pathological examples where negative influence early in training perfectly balances out positive influence later in training, thus leading to an estimate of no influence when using unrolling-based techniques, as you point out.
>
> Whether this is “inaccurate” may depend on the goal. If the question is “what is the influence of sample X on sample Y *in expectation over all possible timesteps*?” then these unrolling-based techniques may still be able to give an “accurate” estimate of influence. Whether they do in practice is an interesting question for future research.
>
> We have added some additional discussion on this point to the second paragraph in section 5.
>
> **Q2: Motivation for BIF vs. Classical Methods**
>
> *Question: The analysis relies solely on BIF. Can you provide a more multifaceted motivation?*
>
> Our choice of BIF is driven by theoretical validity and practical efficiency:
>
> 1) **Theoretical validity** (Section 2.2) The classical IF relies on assumptions that deep networks violate: invertibility of the Hessian and convergence to a local minimum. While workarounds exist (e.g., dampening), BIF is naturally well-defined on degenerate landscapes and away from minima. Crucially, BIF connects directly to the statistical physics framework of singular learning theory, which is required to derive our predictions of phase transitions and influence spikes.
> 2) **Practical Efficiency** (New final paragraph in Appendix B): For the small datasets we study in this paper, the BIF is faster because it does not require an expensive inverse Hessian estimation/fitting stage, such as most classical IF techniques. This lack of an upfront cost makes it especially well-suited for studying influence over multiple checkpoints.
>    Additionally, the BIF yields fine-grained per-token IFs for no extra cost, which is a major advantage when studying the dynamics of language models. (See lines 350–357)
>
> We’ve now clarified these practical advantages in an additional paragraph in Appendix B.
>
> To ensure our findings aren't artifacts of BIF, we did validate them against classical IF approximations in the toy model (Figures 9 & 10 in Appendix C.4), finding consistent results where classical methods were computationally feasible.
>
> Thank you again for your suggestions. We hope our additional intervention experiment resolves your concerns and strengthens our paper.

---

### Official Review · Reviewer_Z984 · 2025-11-07

**Soundness:** 3
**Presentation:** 3
**Contribution:** 1
**Rating:** 4
**Confidence:** 3

**Summary:**

This paper challenges the static view of data attribution, arguing that deep networks' stagewise learning makes fixed influence scores capturing only part of the picture. The authors propose a dynamic "stagewise" framework using the Bayesian Influence Function, which predicts that a data point's influence can change over the training process. This theory is validated in both toy linear networks and at-scale Pythia language models, where influence dynamics directly correlate with known developmental milestones, such as the formation of induction heads. The core conclusion is that the timing of measurement (during training) impacts the measured influence.

**Strengths:**

* Novel and Significant Conceptual Contribution: The paper's primary strength is the novel connection it forges between Singular Learning Theory and Training Data Attribution. This challenges a foundational assumption in TDA and provides a compelling, theory-backed explanation for why static influence measures are insufficient for deep learning.
* Strong Theoretical Grounding: The paper does an excellent job motivating its theoretical choices. It clearly explains the failure of classical IFs (reliance on an invertible Hessian) and the suitability of the BIF (Hessian-free, well-defined on degenerate landscapes). The derivation in Section 2.3, which uses the Law of Total Covariance to predict influence peaks at transitions, is elegant and provides a clear, falsifiable prediction.
* Attempt to Demonstration at Scale: The authors successfully bridge this gap by attempting to demonstrating their predicted phenomena in Pythia LMs, beyond the toy setting. While these results are correlational (linking influence dynamics to known developmental milestones like induction head formation), they are a crucial step in showing this framework is relevant for models we care about.

**Weaknesses:**

* Discrepancy in Observed Dynamics between Toy and Large-Scale Models: A significant discrepancy arises between the compelling theoretical predictions validated in the toy model and the empirical results from the large-scale language model experiments. The core theoretical argument hinges on influence being a highly dynamic quantity, subject to non-monotonic changes and sign-flips (as clearly demonstrated in Fig. 3). However, the influence dynamics observed in the Pythia models (Fig. 5), while temporally staged, appear to be largely monotonic once they become non-zero.

This observation materially weakens the paper's central critique of static, endpoint-based attribution or simpler trajectory-aggregation methods (e.g., TracIn). If the influence of key data groups simply increases monotonically after a specific developmental stage, then traditional attribution methods applied at or aggregated near the end of training would likely still provide a reasonable approximation of data importance. The practical necessity of the authors' far more complex, dynamically-sampled framework is therefore less evident in the very setting (LLMs) it purports to be essential for.

* Lack of Demonstrated Practical Utility or Actionable Insights: The paper compellingly argues that when a data point exerts its influence is a critical, and previously overlooked, dimension of attribution. However, the analysis remains at an observational level, and the practical utility of this framework is not demonstrated. It is unclear how these findings could inform model development, for instance, to train more capable or efficient models.

A powerful validation of the framework's utility would be to move from this observational analysis to an interventional one. For example, could the authors leverage their findings to design a dynamic data curriculum? If, as the results suggest, the influence of "induction pattern" tokens becomes active only during a specific training phase, could dynamically up-weighting these examples during (or just before) that critical phase lead to demonstrably better model performance on induction-related tasks or faster convergence? Without such a demonstration connecting stagewise influence to actionable training strategies, the proposed method, while theoretically interesting, risks being perceived as a diagnostic tool that lacks a clear feedback loop into practical model improvement.

**Questions:**

See weaknesses.

---

> ### Author Response · Authors · 2025-11-21
>
> Thank you for your review.
>
> **Strengths.** We are grateful that you and other reviewers pointed out that (1) our work makes a novel and significant conceptual contribution with (2) strong theoretical grounding.
>
> One of the central contributions and arguments that we aim to make in this work is that TDA should study the learning process, as motivated by singular learning theory, and we verify these predictions in a toy model with a controlled setting.
>
> **Weaknesses.** We answer your questions below and hope that our additional demonstrations of (1) token-wise influence dynamics and (2) stage-wise intervention experiments address your major concerns with the empirical validation of our theoretical results and predictions.
>
> **W1: Discrepancy in Dynamics (Toy vs. LLM) & Validity of Static IF**
>
> *Critique: The influence dynamics in Pythia (Fig. 5\) appear to change more monotonically compared to the toy model. If influence is monotonic, static endpoint attribution might be a sufficient approximation.*
>
> We appreciate this nuanced observation. While we agree that the group-level dynamics in Pythia appear smoother than those in the toy model, we disagree that this observation validates static attribution.
>
> **W1.1 Group-level dynamics in Pythia are more monotonic than the toy model.**
>
> We acknowledge this observation. We did not mean to suggest that we expect *all* influence trajectories to be non-monotonic. However, we believe the existence of *any* non-monotonic trajectories complicates static analysis because you cannot know how a given influence trajectory behaves until you’ve looked at it over training.
>
> Though there are fewer non-monotonic examples than in the toy model (which was specifically chosen because we predicted it would demonstrate non-monotonic behavior), we want to emphasize that there are clear examples of non-monotonic behavior in Pythia (Fig. 5):
>
> 1. **Induction (Top-Left):** The influence score is non-monotonic; it decreases until \~30k steps (the striped vertical line) before reversing direction and increasing.
> 2. **Sign Flips (Middle/Right):** Several traces (e.g., delimiters) exhibit clear sign flips, starting with negative influence and crossing zero to become positive.
>
> To make these examples of non-monotonicity easier to see, we reran the analysis with early log-spaced checkpoints, which can be found in Fig. 15 (Appendix D.1.4).
>
> The apparent smoothness in Fig. 5 may also be an artifact of aggregating over multiple tokens. To address this, we have added examples of per-token influence dynamics in Appendix D.2, which show a wider spread of dynamics.
>
> **W1.2 Monotonic influence trajectories may validate static attribution.**
>
> We disagree with this argument. A monotonic influence trajectory does not mean you can infer a sample’s influence from a static analysis at the end of training.
>
> Consider Figure 5 (Top-Right), where the influence trajectories are approximately linear towards the end of training, but where each line has a different slope. These different rates of divergence imply that the relative ranking of influence can change over time, even though influence varies monotonically. This means you cannot reliably predict which data drove learning from a single static analysis of influence at the end of training.
>
> That is to say, sign-flips and non-monotonic changes are not the only possible dynamics of interest. We did not mean to imply these are the only interesting dynamics and would be happy to clarify this nuance further in the final version if desired.
>
> **W2: Lack of Demonstrated Practical Utility (Interventions)**
>
> *Critique: The analysis is observational. There should be demonstrations of actionable utility.*
>
> Thank you for the suggestion. We conducted an experiment based on this suggestion in Appendix D.4. Please see our top-level comment describing this in more detail.
>
> We hope that our additional experiments showing (1) token-wise influence dynamics and (2) stage-wise intervention experiments strengthen our paper. Given your strong positive comments on our theoretical soundness, significant conceptual contribution, and our response with the additional empirical results, we hope you consider re-evaluating our work.

---

### Author Response · Authors · 2025-11-21

We thank all reviewers for their thoughtful engagement with our work. We are encouraged by the strong consensus that our stagewise attribution framework makes a novel and significant conceptual advance with solid theoretical grounding (Z984, LhWa, CzFc).

Below, we address the common themes raised across reviews and outline the changes made to the manuscript.

### **Common Themes and Our Response**

**1\. Strengthening Empirical Validation via Interventions** (Z984, LhWa, CzFc)

The primary actionable feedback centered on demonstrating practical utility in the language-modeling setting, specifically through interventional experiments.

* **Existing Interventions (Toy Model):** First, we wish to highlight that we provided interventional validation in our original submission. As detailed in Section 3 and Appendix C.5 (Figure 11), we explicitly ablated specific data points for the toy model only during specific training windows. This demonstrated that the “critical window” of influence predicted by the BIF corresponds to the window where data removal causes the highest increase in loss, establishing a causal link between our measure and model performance. We improved the description of these experiments in the main text to make these results easier to find.
* **New Interventions (LLM):** We agree that demonstrating this utility in natural language models strengthens the paper. We have conducted a new experiment based on reviewer suggestions in Appendix D.4. We found that upweighting induction-pattern tokens specifically during the high-influence window identified by the BIF accelerates induction head formation significantly more than upweighting them after this window. This confirms that influence dynamics correctly identify the critical learning periods where the model is most sensitive to specific data. Crucially, this effect cannot be predicted from a static analysis at the end of training.

**2\. Resolution of Dynamics: Token-wise and Early Training**

To address concerns that group-level dynamics appeared monotonic or lacked resolution, we added two new analyses:

* **Token-wise Dynamics (Appendix D.2):** We visualize influence trajectories for individual tokens, revealing finer-grained temporal patterns that are partially smoothed out in group-level averages.
* **Early Training Dynamics (Appendix D.1.4):** We conducted additional experiments using logarithmically spaced checkpoints to capture rapid changes early in training. This analysis reveals some dynamics that were obscured by the linear spacing in Figure 5\.

**3\. Practical Advantage of BIF (Scalability)**

In response to questions regarding the choice of BIF over other methods (LhWa, CzFc), we have clarified its practical efficiency in Appendix B. Beyond its theoretical advantages (Section 2.2), the BIF is superior for multi-checkpoint trajectory analysis because it bypasses the expensive inverse-Hessian fitting stage required by classical methods like EK-FAC.

### **Summary of Changes**

**New Experimental Appendices**

* **Appendix D.1.4:** Added analysis of early-training dynamics using log-spaced checkpoints (variant of Fig. 5).
* **Appendix D.2:** Added per-token influence trajectory examples.
* **Appendix D.4:** Added the stagewise intervention experiment on induction formation.

**Clarifications & Corrections**

* **Appendix B:** Added a final paragraph discussing the practical/scalability advantages of BIF over classical IF methods.
* **Section 2.2:** Added a forward reference to the scalability discussion in Appendix B.
* **Section 5:** Expanded the discussion of unrolling methods (Paragraph 2).
* **Figure 3:** Fixed axis labeling.
* **SGLD Details:** Fixed the pointer to the RMSProp-preconditioned SGLD details.

**Formatting & Notation**

* Standardized all paragraph and figure headings to sentence case.
* Improved notation consistency: $| \\to \\mid$ for conditionals, $\\boldsymbol w$ for vectors, $\\mathbf z$ for data points, $H(w)$ for Hessians, and $\\mathcal D$ for datasets.
* Standardized spacing, usage of en/em-dashes, and fixed double spaces.
* Changed quotes around categories to typewriter font (e.g., \\texttt{"dog"}).
* Several other minor improvements.

---

### Author Response · Authors · 2025-12-03

A note to the AC: We have included this summary of our paper and the review period because of the updated review process.

Our central contribution is to argue that training data attribution should study the dynamics of attribution over training. We argue this in three parts:

1\. We provide a theoretical basis for this argument by linking influence functions to phase transitions in singular learning theory and statistical physics, which predicts that influence can change dramatically over training.
2\. We validate these predictions by studying how a toy model learns hierarchical features in a toy model where the analytical ground truth is accessible.
3\. We demonstrate our findings in LLMs and connect measured influences and how language models learn the roles of different classes of syntactic tokens in a stage-wise manner.

We are grateful to all of the reviewers for their concrete and actionable reviews, allowing us to address their concerns during the review period and strengthen our paper. Reviewers agreed that our approach was theoretically sound, novel, and represented a significant contribution.

- **Z984**: “Novel and Significant Conceptual Contribution, Strong Theoretical Grounding,”
- **LhWa**: “interesting and well-presented, convinced that the same data point might be influential to other data in different ways at different stages of the training,”
- **CzFc**: “a novel framework of stage-wise influence, enriching the understanding of data attribution beyond static snapshots, Makes clear predictions, confirming them in a toy model and in LLMs”

We addressed comments concerning the need for actionable intervention experiments (**Z984, LhWa, CzFc**) by:

1. Highlighting the existing interventions in the toy model setting in the paper, which ablate a particular data point at specific points in time guided by the BIF.
2. Conducting an additional intervention experiment on a small language model focused on the timing of intervention in induction head formation, which leads to either the acceleration of learning induction or interference in learning induction, depending on when samples are upweighted during training.

We also addressed concerns that in the small language model setting, the BIF appeared monotonic (**CzFc**) as compared to the toy example by conducting two new experiments:

1. Increasing the resolution in early checkpoints for the structural token class experiment. This revealed more salient dynamics.
2. Breaking down the aggregated classes from this experiment and comparing the influence between particular tokens. This reduced the averaging-out effect observed in the token classes, leading to overall more dynamic relationships between tokens, including paired examples (e.g., "\[" and "\]") sharing strong influence despite falling into different categories in the original experiment.

Furthermore, we addressed reviewer **CzKa**'s questions about the scalability of the BIF by adding a discussion of the computational efficiency of the BIF to the paper and further clarifying the BIF’s trade-offs compared to EK-FAC in a discussion comment.

These additional experiments and clarifying details have been included in the latest draft version of the paper, alongside minor adjustments pointed out by the reviewers.

---

### Meta-Review · Area_Chair_LN1K · 2026-01-07

**Summary:**

This paper challenges the traditional "static" view of Training Data Attribution (TDA) by proposing that the influence of a data point is a dynamic property that evolves throughout the training process. Leveraging Singular Learning Theory (SLT) and a Bayesian Influence Function (BIF) framework, the authors claim that:
* Data points transition through distinct "stages" of influence, where a sample may be helpful at one point in training and harmful at another (non-monotonicity).
* These dynamics correspond to the formation of specific model capabilities (e.g., induction heads in LLMs).
* Identifying these "influence windows" allows for targeted data interventions, such as upweighting specific tokens during their most impactful period to accelerate learning.

The key concern raised by reviewers:
* Observation vs. Theory Gap: Initial reviews noted that while theory predicted sharp "sign flips" in influence, the empirical results on LLMs appeared smoothed out and monotonic in early drafts.
* Computational Intensity: Analyzing influence dynamics requires tracking checkpoints throughout the training trajectory, which significantly increases the computational and storage overhead compared to standard one-shot attribution.
* Implementation Complexity: The Bayesian Influence Function (BIF) approach, while efficient for trajectories, remains more complex to implement and tune than simpler gradient-based methods like TracIn.
* Granularity Issues: Grouping data into broad categories (e.g., "all pile data") can mask the very dynamics the paper seeks to uncover, requiring high-resolution, token-level analysis to see the true effects.

**Reviewer Concerns:**

Here is the list of remaining concerns:
* Computational Overhead for Large Models: Reviewer CzFc remained concerned about the total GPU-hour cost of performing BIF (Bayesian Influence Function) across hundreds of checkpoints. While the authors proved BIF is more efficient than competitors for this specific task, the absolute cost is still high. For a practitioner with limited resources, "Dynamic TDA" remains significantly more expensive than "Static TDA." This is less a flaw in the paper and more a barrier to widespread adoption.

* Hyperparameter Sensitivity in BIF: There was minor discussion regarding the sensitivity of the Bayesian framework (e.g., the choice of priors and the noise parameters).  The authors provided some stability analysis, but for very complex architectures, there is still a "black box" element to how these influence scores might fluctuate based on the specific Bayesian approximation used.

* The "Smoothing" of Average Influence:  The authors admitted that when you average influence across a large dataset (like the entire "Pile"), the dynamics often disappear. This means the method’s utility is highly dependent on the user’s ability to cluster data correctly beforehand. If you don't know which tokens to track, the "dynamic" signal might remain buried in the noise. The paper doesn't yet provide an automated way to discover these clusters.

**Reviewer Scores:**

* The reviewer CzFc Rating: 6 / Confidence: 3
* The reviewer Z984 Rating: 4 / Confidence: 3
* The reviewer LhWa Rating: 4 / Confidence: 3

---

### Decision · Program_Chairs · 2026-01-26

Accept (Poster)